# QTL analysis of femaleness in monoecious spinach and fine mapping of a major QTL using an updated version of chromosome-scale pseudomolecules

Kaoru Yamano[1☉], Akane Haseda[1☉], Keisuke Iwabuchi[1☉], Takayuki Osabe[2☉], Yuki Sudo[1☉], Babil Pachakkil[3‡], Keisuke Tanaka[4,5‡], Yutaka Suzuki[6‡], Atsushi Toyoda[7‡], Hideki Hirakawa[8]*, Yasuyuki Onodera[9]*

1 Graduate School of Agriculture, Hokkaido University, Sapporo, Japan, 2 School of Agriculture, Hokkaido University, Sapporo, Japan, 3 Department of International Agricultural Development, Faculty of International Agriculture and Food Studies, Tokyo University of Agriculture, Setagaya-ku, Tokyo, Japan, 4 NODAI Genome Research Center, Tokyo University of Agriculture, Setagaya-ku, Tokyo, Japan, 5 Department of Informatics, Tokyo University of Information Sciences, Chiba, Japan, 6 Department of Computational Biology and Medical Sciences, Graduate School of Frontier Sciences, The University of Tokyo, Kashiwa, Japan, 7 Department of Genomics and Evolutionary Biology, National Institute of Genetics, Mishima, Japan, 8 The Department of Technology Development, Kazusa DNA Research Institute, Kisarazu, Japan, 9 The Research Faculty of Agriculture, Hokkaido University, Sapporo, Japan

☉ These authors contributed equally to this work.
‡ BP, KT, YS and AT also contributed equally to this work.
* onodera@agr.hokudai.ac.jp (YO); hh@kazusa.or.jp (HH)

**Data Availability Statement:** All NGS data are available from DDBJ Sequence Read Archive

## Abstract

Although spinach is predominantly dioecious, monoecious plants with varying proportions of female and male flowers are also present. Recently, monoecious inbred lines with highly female and male conditions have been preferentially used as parents for $F_1$-hybrids, rather than dioecious lines. Accordingly, identifying the loci for monoecism is an important issue for spinach breeding. We here used long-read sequencing and Hi-C technology to construct SOL_r2.0_pseudomolecule, a set of six pseudomolecules of spinach chromosomes (total length: 879.2 Mb; BUSCO complete 97.0%) that are longer and more genetically complete than our previous version of pseudomolecules (688.0 Mb; 81.5%). Three QTLs, *qFem2.1*, *qFem3.1*, and *qFem6.1*, responsible for monoecism were mapped to SOL_r2.0_pseudomolecule. *qFem3.1* had the highest LOD score and corresponded to the *M* locus, which was previously identified as a determinant of monoecious expression, by genetic analysis of progeny from female and monoecious plants. The other QTLs were shown to modulate the ratio of female to male flowers in monoecious plants harboring a dominant allele of the *M* gene. Our findings will enable breeders to efficiently produce highly female- and male-monoecious parental lines for $F_1$-hybrids by pyramiding the three QTLs. Through fine-mapping, we narrowed the candidate region for the *M* locus to a 19.5 kb interval containing three protein-coding genes and one long non-coding RNA gene. Among them, only *RADIALIS-like-2a* showed a higher expression in the reproductive organs, suggesting that it might play a role in reproductive organogenesis. However, there is no evidence that it is involved in the regulation of stamen and pistil initiation, which are directly related to the floral sex

(accession numbers DRX241308, DRX477737, DRX241317–DRX241420, DRX094096–DRX094113, DRX477738–DRX477746). All genome and gene sequences and annotations are available at figshare (https://doi.org/10.6084/m9.figshare.23916627.v1).

**Funding:** This work was supported by the Japan Society for the Promotion of Science (Grants-in-Aid for Scientific Research [KAKENHI] Grant Number 21K05525 and 22H02321), Ministry of Education, Culture, Sports, Science and Technology (KAKENHI Grant Numbers 16H06279). The funders had no role in study design, data collection and analysis, decision to publish, or preparation of the manuscript.

**Competing interests:** The authors have declared that no competing interests exist.

differentiation system in spinach. Given that auxin is involved in reproductive organ formation in many plant species, genes related to auxin transport/response, in addition to floral organ formation, were identified as candidates for regulators of floral sex-differentiation from *qFem2.1* and *qFem6.1*.

## Introduction

Originating from Persia, spinach has since been introduced widely across East Asia, Europe, and the Americas. Spinach is now grown as a nutritious vegetable rich in vitamins, dietary fiber, minerals, and antioxidants such as carotenoids and flavonoids, and is produced in more than 50 countries around the world, with a total annual production of 32,300 kt [1, 2]. Spinach is commonly known as a dioecious species consisting of male and female individuals. However, certain varieties, lines, germplasm accessions, and crosses have been shown to produce monoecious plants, which bear both female and male flowers in various proportions in the same individual [3]. Most of the modern spinach varieties for commercial production are $F_1$-hybrids. In the early stages of spinach hybrid breeding, the promising combinations of dioecious lines were planted in alternating rows with male plants removed before anther dehiscence from the seed-parental rows. Later, to save labor by eliminating the requirement of removing the males, monoecious lines breeding true for a highly female condition began to be used as seed parents for hybrid seed production [4, 5]. Currently, highly male monoecious lines are also frequently used as pollen parents in spinach breeding. Using monoecious lines reduces the labor required to remove male plants from the seed-parental rows. Moreover, highly male monoecious plants have a longer lifespan after flowering than male plants, making it easier to synchronize pollination timing with the maternal (seed-parental) line (personal communication from Tohoku Seed Co.). Therefore, understanding the mechanisms of sex determination in dioecious lines and controlling the degree of femaleness in monoecious spinach are critical issues in hybrid breeding programs.

Most flowering plants have both male and female reproductive organs—i.e., they are hermaphroditic or monoecious—but some have separate sexes. Dioecious plants, which have separate sexes, are taxonomically widely dispersed although they are found in only a small percentage of flowering plants [6]. This suggests that having both sexes in one plant (hermaphroditism or monoecism) is the ancestral condition and that dioecy has arisen recently and independently in many evolutionary lineages. The sex of dioecious plants is often determined by genetics, and sex chromosomes have been found in a wide range of angiosperm lineages. Therefore, given the evolutionary history of dioecy, it is plausible that plant sex chromosomes also have evolved recently from autosomes in their hermaphroditic or monoecious ancestors [7, 8]. Some dioecious plants, such as white campion, sorrel, and hemp, have heteromorphic sex chromosome pairs, while others, such as spinach, asparagus, and papaya, have homomorphic sex chromosome pairs. Sex chromosome pairs are derived from autosomal pairs, strongly suggesting the evolutionary transition from homomorphic to heteromorphic sex chromosome pairs. Thus, homomorphic and heteromorphic sex chromosomes that have emerged in various angiosperm lineages may represent different evolutionary stages. In dioecious spinach lines, the sexual dimorphism is controlled by an allelic pair designated *X* and *Y*, which are located on the longest chromosome pair [9]. Specifically, the chromosome with the dominant male determinant is the Y chromosome, while its counterpart is the X chromosome [10, 11]. The spinach sex chromosomes are cytogenetically homomorphic with some exceptions and recombine across most of their length [12]. However, recombination of the Y with the X chromosome is

absent in the region surrounding the male-determining locus [13]. A recent study predicted that the non-recombining region of the Y chromosome is 17.4 Mb in length, accounting for ~8% of the total length [14].

In our previous genetic analysis using progeny families from the cross of dioecious line 03–009 with the highly male monoecious line 03–336, a single, incompletely dominant gene designated *M* was shown to be responsible for the monoecious character. The monoecious gene (*M*) is linked to the male-determining locus with a distance of ~13 cM, located at a 7.5 cM chromosomal region bracketed by variant sites at *clc-e* and *clsy3* loci. Furthermore, the genetic analysis of progeny from the cross between female and monoecious plants suggested that the male-determining gene (*Y*) is epistatic to the monoecious gene (*M*), which is epistatic to *X*: *XXmm* is female, *XYMM*, *XYMm* and *XYmm* are male, and *XXMM* and *XXMm* produce a monoecious phenotype [15–17]. Based on the phenotypic expression of plants possessing these genotypes, it can be inferred that *Y* and *M* act as male-promoting and female-suppressing factors, with the effect of *M* being weaker than that of *Y*.

Apart from the major factor that determines the development of monoecism, there are modifying genes that influence the character of the monoecism, and these are inherited quantitatively. It has been observed that selection can produce monoecious lines that have different levels of femaleness, as measured by the proportion of female to male flowers per plant. Through three generations of inbreeding, Janick and Stevenson succeeded in selecting plants with progeny mean values that ranged from 70–80% female to as low as 5% female [18]. However, the additional loci for monoecism have not yet been identified.

Having high-quality genome assemblies, such as chromosome-scale pseudomolecules, is crucial for identifying genes and loci of interest. Such pseudomolecules are particularly important as a reference because GWAS and QTL analyses require highly accurate variant data that covers nearly the whole genome. The pseudomolecules can be constructed by aligning highly accurate scaffolds in the appropriate direction and have high genome coverage rates. Furthermore, chromosome-scale pseudomolecules lay the foundation for genome-based molecular breeding strategies, such as marker-assisted and genomic selection [19].

The spinach genome consists of $n = x = 6$ chromosomes and is estimated to be 989 Mb, based on the C-value. Recently, a genome assembly (SOL_r1.1) for the dioecious spinach line 03–009 was generated by single-molecule sequencing technologies (e.g., PacBio, ONT, and 10x Genomics) in combination with mate-pair reads [20]. The spinach genome assembly represented >90% of the whole genome and showed high continuity ($N_{50} = 11.3$ Mb), but it contained chimeric scaffolds caused by assembling errors. Finally, only ~70% of the genome assembly was predicted to be correctly assembled, and this portion was anchored to a set of six pseudomolecules that were collectively designated SOL_r1.0_pseudomolecule [20]. Using pseudomolecules with a genome coverage rate of about 70% as a reference for GWAS and QTL analyses, there is a risk that the genes of interest may not be detected and identified. Therefore, in this study, we enhanced the length and accuracy of the pseudomolecules using Hi-C, and mapped three QTLs for controlling femaleness in monoecious spinach to the updated pseudomolecules. In addition, one of the QTLs, corresponding to the *M* locus, was fine-mapped to a 19.5 kb region, and transcriptome analysis was carried out to elucidate the underlying mechanisms controlling femaleness in monoecious spinach.

## Materials and methods

### Plant materials

In this study, to identify the loci controlling femaleness in monoecious plants, we conducted QTL analyses using female plants from the dioecious line 03–009 and plants from the highly

male monoecious line 03–336 with ~5% femaleness scores as parental plants for producing the mapping population.

Line 03–009 produced strict female (100% femaleness [proportion of female flowers per plant]) and male (0% femaleness) plants, while line 03–336 showed the lowest femaleness scores among monoecious lines stored in our lab [3]. The spinach $F_2$ population (03–009 x 03–336 $F_2$; [20]) resulted from crossing a female plant ($mm$ plant) of line 03–009 with a plant ($MM$ plant) from the highly male monoecious line 03–336 and was used for linkage map construction and QTL analysis. $S_1BC_2F_1$–$S_5BC_2F_1$, $F_3$–$F_4$, and $BC_1F_4$ generations from the cross between 03–009 and 03–336 were used for fine mapping of the $M$ locus (S1 and S2 Figs) [3]. Furthermore, the plants homozygous for $M$ ($MM$ plants) selected from the $F_2$ and $F_3$ generations of the 03–009 × 03–336 cross were used for the verification of minor QTLs for the femaleness phenotype (the percentage of female flowers per plant) in monoecious plants. The genomic region carrying the monoecious ($M$) gene was introgressed by five successive backcrosses from monoecious donor line 03–336 into the 03–009 genetic background. This produced the near-isogenic line NIL-M (S3 Fig). By using lines NIL-M and 03–336 we anticipated that we could clarify the influence of the different genetic backgrounds on monoecious expression by the $M$ gene.

For preliminary RNA-seq, RT-qPCR, and QTL analyses, plants were cultivated in a growth chamber (LH-350S; Nippon Medical & Chemical Instruments, Osaka, Japan) at 20˚C with an 8 h photoperiod for the first four weeks, at 23˚C with an 8 h photoperiod for the next three weeks, and subsequently at 25˚C with a 24 h photoperiod. For QTL verification and transcriptome analysis, plants were maintained at a consistent 20˚C, with an 8 h photoperiod for the first four weeks, a 16 h photoperiod for the next two weeks, and finally a 24 h photoperiod. For this study, we evaluated the femaleness of plants that bore over 20 flower clusters, with each cluster comprising 5–10 flowers. For expression analysis, vegetative leaves, shoot apexes (plants under short-day conditions, and plants with 3 or 7 days of long-day exposure after being grown in short-day conditions), and early-stage (~5 mm length) and mid-stage (30–40 mm length) inflorescences were harvested (S4 Fig). Total RNA was extracted using an RNeasy Plant Mini Kit combined with an RNase-Free DNase Set (QIAGEN, Venlo, Netherlands). Total cellular genomic DNA was prepared using the method of Sassa [21].

## Polishing and scaffolding of the spinach *de novo* genome assembly

A phased long-read-based genome assembly, Spol_r0.0 ([20]; S1 Table), of a male plant of dioecious line 03–009 was polished by Racon [22] with long-reads (DRX241314; [20]) and by Pilon [23] with Illumina PE reads (DRX241308). Hi-C analysis was used because it produces pairs of short reads formed by crosslinking chromatin interactions. In addition, Hi-C analysis provides genomic proximity information with a scale ranging from 1 kb to 1 Mb or longer, which allows for the accurate assignment, ordering, and orientation of genomic sequences [24]. To assemble Spol_r0.0 into a set of more continuous genome sequences by scaffolding with Hi-C data, the Hi-C library was prepared from young leaves harvested from a single male plant of line 03–009 by using an Arima Hi-C Kit (Arima Genomics, San Diego, CA) and sequenced by a HiSeq X Instrument (Illumina, San Diego, CA) configured to generate 151-bp PE reads (DRX477737; S2 Table). Scaffolding of contigs was carried out using the Hi-C mate pairs with SALSA [25]. Illumina short reads obtained from a single male and a single female plant (DRX241308 and DRX241308; [20]) were mapped to Spol_r0.0, and calculated the male-to-female read depth ratio (M/F ratio) on assembly contigs and used to identify sex-linked (Y-linked) regions: the Y-linked region tends to have a higher male (XY) read depth than female (XX) read depth, as it does not recombine with the X chromosome and tends to be divergent from the X-linked

region. In this study, contigs with $\log_{10}$(M/F ratio) > 0.3 were considered Y-linked. The M/F ratio scores were used to specify chimeric scaffolds composed of contigs representing X- and Y-linked regions and to filter out Y-linked regions from the chimeric scaffolds.

## Construction of linkage map and QTL analysis

The double-digest restriction site-associated DNA sequencing (ddRAD-Seq) reads (DRX241317–DRX241420) obtained from 101 $F_2$ plants and an $F_1$ plant from a 03–009 × 03–336 cross and their parents (03–009 and 03–336) were mapped to the spinach draft genome (Spol_r1.0; S1 Table). SNPs were called as described in Hirakawa et al. [20]. The raw SNPs were filtered by using VCFtools [26] and SnpEff [27]. Construction of the linkage map was carried out by using the Lep-Map3 software package [28]. QTL analysis was performed using the CIM function of the R/qtl package [29] with the Haley–Knott regression method [30]. The LOD significance threshold for detecting QTLs was calculated by performing 1,000 iterations using the R/qtl permutation test. ANOVA was used to calculate the phenotypic variance explained (PVE) for each QTL.

## Construction of a linkage map and pseudomolecules

Scaffolds of the spinach draft genome were anchored to the SNP linkage groups and assembled into draft pseudomolecules. The Hi-C mate pairs obtained from the 03–009 were mapped to the draft pseudomolecules, and Hi-C contact maps were built by using the software Juicer 1.6 [31]. In accordance with to the Hi-C contact maps, the draft pseudomolecules were manually corrected to obtain the final version of the spinach pseudomolecules. The pseudomolecules were annotated with 29,276 genes (SOL_r1.1a; [20]) using the software GeMoMa 1.9 [32]. The quality of the assembled genome and predicted genes was assessed by Benchmarking Universal Single-Copy Orthologs (BUSCOs). The BUSCO completeness scores of the 1416 single-copy orthologous genes defined in embryophuta_odb10 dataset were calculated by BUSCO v5.3.1 [33].

## Detection and annotation of nucleotide variants

Whole-genome sequencing (WGS) libraries for the spinach lines 03–009 and 03–336 were prepared as described in Hirakawa et al. [20] and sequenced using a HiSeq 2500 Instrument (Illumina) configured to generate 251-bp pair-end reads. The WGS reads (DRX401201, DRX401202; S3 Table) were mapped to the spinach pseudomolecules obtained herein using the software Bowtie2 Version 2.5.1 [34], and nucleotide variants were called. The impacts of nucleotide variants on gene function were evaluated using SnpEff 5.0e software [27] and classified as 'HIGH', 'MODERATE', 'LOW' or 'MODIFIER'. A variant categorized as HIGH is expected to have a significant disruptive impact on the protein. This may lead to the truncation of the protein, loss of function, or trigger nonsense-mediated decay. A variant categorized as MODERATE is not expected to cause significant disruption but may alter protein effectiveness. A variant categorized as LOW is considered to be mostly harmless or unlikely to change the behavior of the protein. Finally, the fourth category, MODIFIER, is reserved for non-coding variants or variants affecting non-coding genes, where predictions are difficult or there is no evidence of impact.

## DNA marker analysis

Genetic markers for identifying allelic variants are listed in S4 and S5 Tables. Except for Sequence Characterized Amplified Region (SCAR) markers, all markers used in this study were developed by cleaved amplified polymorphic sequence (CAPS) and derived cleaved

amplified polymorphic sequence (dCAPS) marker techniques [35]. Amplification and detection of the markers were performed as described previously [17].

## Reverse transcription quantitative-PCR

The cDNA used as a template for reverse transcription quantitative-PCR (RT-qPCR) was synthesized according to the manufacturer's instructions using SuperScript® IV Reverse Transcriptase (Invitrogen). qPCR was performed with at least three biological replicates, each with two technical replicates, using PowerUp SYBR Green Master Mix (ThermoFisher). Quantification of the expression levels of the genes of interest was carried out using the comparative critical threshold ($\Delta\Delta$Ct) method normalized with *Actin-7* mRNA. The primers used for the RT-qPCR are shown in S6 Table.

## *In situ* hybridization

*In situ* hybridization techniques were used to determine the spatial and temporal expression pattern of *SoRL2a* during flower development. A 637-bp *SoRL2a* cDNA fragment was cloned into pBluescript II SK vector, and *in vitro* transcribed to generate sense and antisense riboprobes. The riboprobes were labeled with digoxigenin using a DIG RNA Labeling Kit (Roche Diagnostics GmbH, Mannheim, Germany). Plant tissues were infiltrated with 10% formalin neutral buffer solution (FUJIFILM Wako Pure Chemical Corporation, Osaka, Japan) at 4˚C for 2 days. They were then dehydrated, embedded in Paraplast and sectioned at 5 μm. The tissue sections were subjected to *in situ* hybridization using an ISH Reagent kit (Genostaff, Tokyo) according to the manufacturer's instructions, and were counterstained with Kernechtrot stain solution (Muto, Tokyo). DIG–labeled riboprobes were detected with anti-Digoxigenin AP Conjugate (Roche Diagnostics GmbH) and visualized using NBT/BCIP Solution (Sigma, St. Louis, MO). Tissue sections were also stained with hematoxylin-eosin to check their morphology.

## RNA-seq-based transcriptome analysis

Using the method described in Takahata et al. [16], we prepared RNA seq libraries from the RNA samples for preliminary expression analysis and sequenced using a HiSeq 2500 Instrument (Illumina) configured to generate 100-bp paired-end reads (DRX094096–DRX094113; S7 Table). The RNA-Seq reads for each RNA sample were quality filtered and adaptor trimmed, and separately mapped to the spinach genome assembly SOL_r1.1 (http://spinach.kazusa.or.jp; [20]) using HISAT 2.2.1 (https://github.com/DaehwanKimLab/hisat2; [36]). Expression values of genes were estimated using StringTie and Ballgown software [37, 38].

For the primary expression analysis, we prepared RNA seq libraries using the NEBNext Ultra II Directional RNA Library Prep Kit (New England Biolabs, Beverly, MA) according to the manufacturer's protocol. The library mixture was sequenced by $1 \times 100$-bp single read sequencing using the Illumina platform NextSeq 1000 (Illumina). The read data were deposited in the DDBJ Sequence Read Archive (DRX477738–DRX477746 S7 Table). Differentially expressed gene (DEG) analysis was performed using a CLC Genomics Workbench 22 (Qiagen, Hilden, Germany). The clean reads were mapped to the reference genome. After statistical analysis based on a generalized linear model, DEGs with a change of more than |2|-fold and a false discovery rate (FDR)-adjusted p-value < 0.05 were selected.

## GO enrichment analysis

The genes predicted in SOL_r1.1a [20] were annotated with Gene Ontology (GO) terms using eggNOG mapper [39]. To elucidate the biological and physiological meaning of the

differentially expressed genes (DEGs) identified in this study, we carried out GO enrichment to identify GO terms over-represented in the DEGs. The significance of the enrichment of GO terms with DEGs was evaluated using Fisher's exact test implemented in the R package topGO [40]. In addition to the conventional Fisher's exact test (classic), the statistical significance was assessed by three other algorithms, i.e., elim, weight, and weight01, to identify significantly enriched GO terms for biological processes. In this study, a GO term was considered significantly enriched when the p-value calculated with the elim, weight, or weight01 algorithms was less than 0.01.

## Results

### Construction of a linkage map and chromosome-scale pseudomolecules for spinach line 03–009

To obtain accurate variant data covering the whole genome required for linkage and QTL analyses, we set out to create a spinach reference genome by correcting pseudomolecules (SOL_r1.0_pseudomolecule) for spinach line 03–009. To achieve this, the initial genome assembly Spol_r0.0 generated by Hirakawa et al. [20] was refined using both long and short reads and then scaffolded using Hi-C mate-pair reads (S1 and S2 Tables). As shown in S1 Table, the resultant genome assembly Spol_r1.0 consists of 323 scaffolds and has a total length of 935.4 Mb with higher continuity (N50 = 74.6 Mb) and BUSCO completeness (97.4%) in comparison to Spol_r0.0. This finding is consistent with a recent study which showed that assemblies with relatively high N50 values consistently had high BUSCO scores [41].

To construct a linkage map, ddRAD-seq reads obtained from $F_2$ plants and an $F_1$ plant from a 03–009 × 03–336 cross and their parents (03–009 and 03–336) were mapped to Spol_r1.0, and 171,522 SNPs were called from this assembly. Linkage analysis of 1816 filtered SNPs produced six linkage groups ranging from 62.2 to 106.6 cM with a total size of 468.6 cM (Fig 1, S8 Table). Since the ddRAD-seq reads were derived from female (03–009) and monoecious (03–336) plants having XX sex chromosome type, one of the six linkage groups—i.e., the group that was sex-linked—was considered to represent the X chromosome. Therefore, although the genome assembly Spol_r1.0 was derived from genomic DNA of the male plant having XY sex chromosome type, we intended to construct pseudomolecules representing a haploid set containing the X chromosome by anchoring Spol_r1.0 to the six linkage groups. The 65 scaffolds (893.1 Mb in total) accounting for 95.5% of the total length of Spol_r1.0 were anchored to the six linkage groups. As five of the anchored scaffolds were found to contain segments (13.9 Mb in total) derived from Y-specific non-recombining regions, based on the M/F ratio score (see the Materials and Methods section for details), the scaffolds were trimmed by filtering out the segments. The trimmed scaffolds (879.2 Mb in total), accounting for 94.0% of the total length of Spol_r1.0, were oriented and ordered with reference to the linkage map, were catenated while filling in gaps between them with Ns of 10 kb in length, and were built into a provisional set of six pseudomolecules. Hi-C contact maps of the pseudomolecules were constructed to detect errors caused by wrong ordering and orientation of the scaffolds. The final set of pseudomolecules, SOL_r2.0_pseudomolecule, was produced by several rounds of manual error correction guided by the Hi-C contact map (S8 Table). As shown in S5 Fig, SOL_r2.0_pseudomolecule was validated using its Hi-C contact map, which suggested that the scaffolds were mostly properly integrated into the final set of the pseudomolecules. Based on the BUSCO methodology [33], the pseudomolecules were assessed as representing 97.0% of genome assembly completeness. Furthermore, based on the LTR Assembly Index (LAI) [42], which evaluates assembly continuity using long terminal repeat retrotransposons (LTR-RTs), SOL_r2.0_pseudomolecule was shown to be of reference grade quality (10 < LAI) (S1 Table).

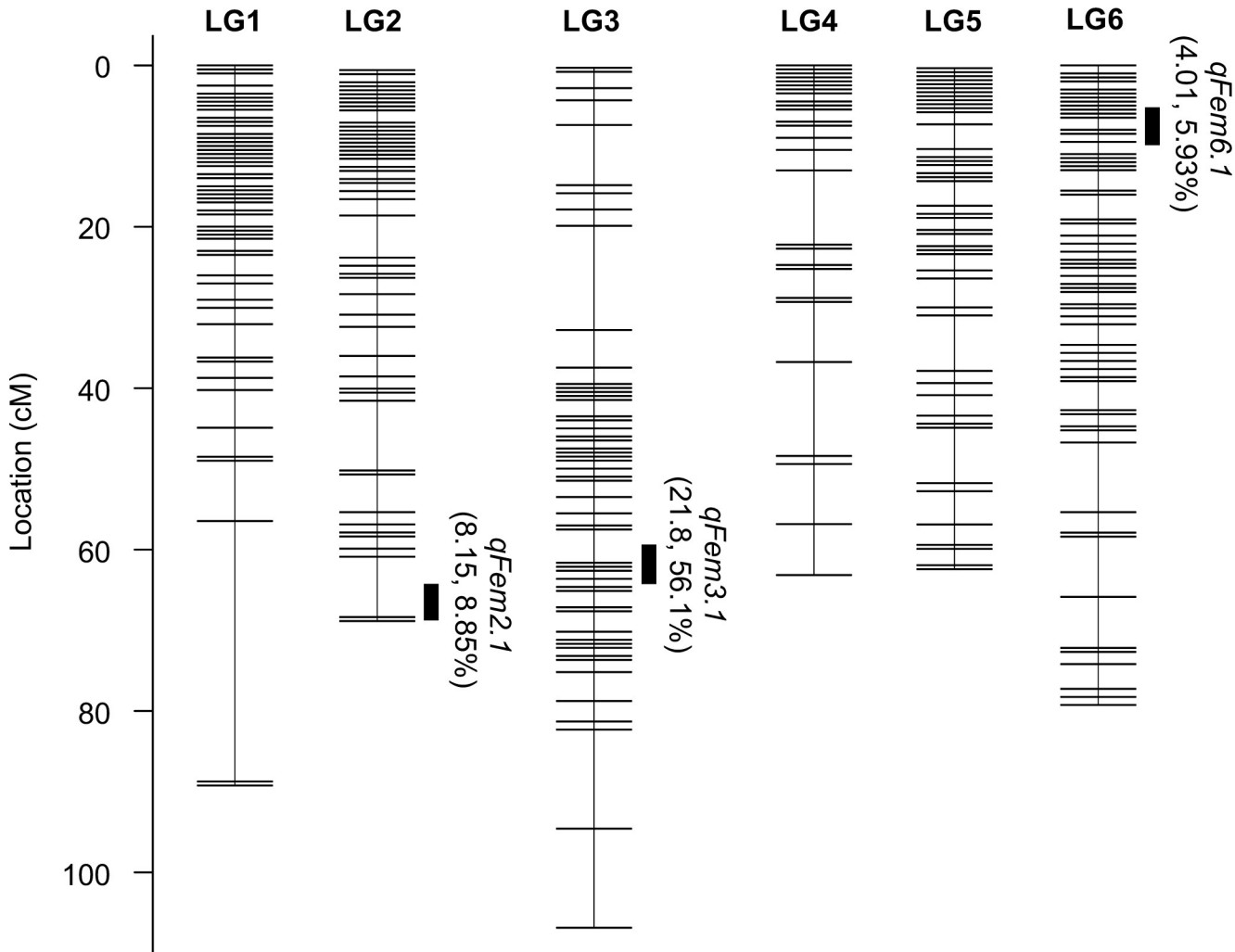

**Fig 1. Molecular linkage map of spinach based on the mapping population 03–009 x 03–336 F₂.** Black bars on the right side of the map represent QTLs (95% CI) for femaleness (proportion of female flowers per plant). The left and right values in parentheses represent LOD values and phenotypic variance explained (PVE) values of the QTLs.

It may be worth noting that the longest pseudomolecule (Chr3), which corresponded to linkage group LG3, was found to represent the sex chromosomes. This is because the protein-coding genes (*phosphoglucomutase* [*pgm*]/g22838.t1 [Chr3_g077230.1], *acyl-activating enzyme 6* [*aae6*]/g17189.t1 [Chr3_g069500.1], *aminotransferase* [*tat2*]/g27450.t3 [Chr3_g042770.1], *nitrite reductase* [*nir*]/g9296.t1 [Chr3_g027530.1], and *chloride channel protein* [*clc-e*]/g49095.t1 [Chr3_g017620.1]) that were previously mapped on the sex chromosomes were found on this longest pseudomolecule (DOI: 10.6084/m9.figshare.23916627) [16].

### Fine mapping of the *M* locus and expression analysis of its candidate genes

In our previous study, the locus responsible for monoecism (*M*) was identified as a major factor responsible for the expression of the monoecious character, and positioned between the *clc-e* (g49095.t1 [Chr3_g017620.1]; Chr3 21.9 Mb) and *clsy3* (g48854.t1 [Chr3_g019780.1]; Chr3 24.7 Mb) loci corresponding to the two DNA markers SP_0008 and SP_0140 on the sex chromosome. Based on the spinach reference genomes (SOL_r2.0_pseudomolecule; S8 Table),

the interval between *clc-e* and *clsy3* loci was estimated to be 2.8 Mb in length (S6 Fig). Here, we attempted to further narrow down the *M* candidate region in order to identify the DNA sequence corresponding to the *M* gene. As in our previous study, progeny families ($S_3BC_2F_1$ generation) from a monoecious selection ($S_2BC_2F_1$−34–1; Fig 2 in Takahata et al. [16]) in the 03-009x03-336 $S_2BC_2F_1$ population were examined to narrow down the candidate region for the *M* locus, using seven DNA markers spanning the interval of *clc-e* and *clsy3* (S6 Fig).

As shown in S6 Fig, $S_2BC_2F_1$−34–1 is heterozygous at the interval of SP_0008 and SP_0140 except at the SP_0008 and SP_0124 locus, which is homozygous for the 03–009 and 03–336 genotypes, respectively. Self-pollination of $S_2BC_2F_1$−34–1 produced monoecious and female $S_3BC_2F_1$ plants, suggesting that the interval is heterozygous for the monoecious gene. Among the five markers bracketed by SP_0008 and SP_0140, only the SP_0036 genotypes co-segregated with the monoecious phenotype in the 15 $S_3BC_2F_1$ plants examined, suggesting that the *M* locus is located around the marker locus bracketed by SP_0124 and SP_0054 (S6 Fig).

For fine mapping of the *M* locus, progeny families from monoecious selections ($S_3BC_2F_1$−34-1-24 and $F_3$−1217-5-116-12) in 03-009x03-336 $S_3BC_2F_1$ and in $F_3$ populations were analyzed using nine markers spanning from SP_0124 to SP_0054 (S7 Fig). As shown in S7 Fig, five markers spanning from SP_0138 to SP_0168 were found to be co-segregated with the monoecious phenotype in the nine progeny plants from $S_3BC_2F_1$−34-1-24, while three markers spanning from SP_0036 to SP_0054/SP_0053 were co-segregated with the phenotype in the three progeny plants from $F_3$−1217-5-116-12. The results from the analysis of the progeny families indicate that the locus for monoecism (*M*) is located in the interval between SP_0168 and SP_0036, which comprises a 20.7-kb physical distance.

To verify the location of the *M* locus, self-pollinated progeny ($S_5BC_2F_1$ and $S_6BC_2F_1$ generations) from the monoecious selection "34-1-24-14" in 03-009x03-336 $S_4BC_2F_1$ generation, and backcross progeny ($BC_1F_4$ and $BC_2F_4$) from the female selection "1217-5-116-12-54" in the 03-009x03-336 $F_4$ generation were examined (S7 Fig). As shown in Fig 2, phenotypes and genotypes of the progeny plants supported the notion that the *M* locus locates at the interval SP_0168–SP_0036. The chromosomal recombination sites in the interval of the progeny plants were determined by sequencing, which further narrowed down the *M* candidate region to a 19.5 kb interval between SNP1 and SNP30 (Fig 2).

As shown in Fig 2C and Table 1, three protein-coding genes, namely, *RADIALIS-like 2a* (*SoRL2a*/g48915.t1 [Chr3_g019240.1]), an uncharacterized protein coding gene (g48913.t1 [Chr3_g019250.1]) and a gene for Acyl-coA acyltransferase-related enzyme 2 required for viability (*SoARV1*/g48912.t1 [Chr3_g019260.1]), were predicted in the 19.5-kb *M* candidate region. It is worth mentioning that the *M* candidate region contains short DNA fragments having high homology with homologs for two Arabidopsis B-class floral organ identity genes, spinach *PISTILLATA* (*SpPI*) and *APETALA3* (*SpAP3*) (GQ120477 and GQ120478): the 343-bp and 97-bp sequences in the *M* candidate region were found to have high homology (89% and 73% identity) with the 5th and 3rd introns of *SpPI*, respectively (Fig 2C). The 343-bp sequence also shows high homology (85% identity) to the upstream region of *SpAP3*. Furthermore, the 75-bp sequence in the *M* candidate region was found to have high homology (90% identity) to the 6th intron of *SpAP3*.

Sequence comparison of BAC clones carrying the *M* locus of lines 03–009 and 03–336 led us to find 45 variant sites in the 19.5-kb region (30 SNPs, 9 indels, and 2 structural variants [SV]) (Fig 2C). However, these variants were not located in coding regions of the three protein-coding genes (Fig 2C), though three variants (1 indel, 2 SNPs) were found in the introns of *SoARV1* and *SoARV1-2*. By contrast, a 7-bp indel and a 327-bp indel (SV) were found in an exon and an intron of the *M-lncRNA* locus, respectively. Two SNPs and one indel were found within the 500-bp upstream flanking region of the *lncRNA* locus. Furthermore, three SNPs

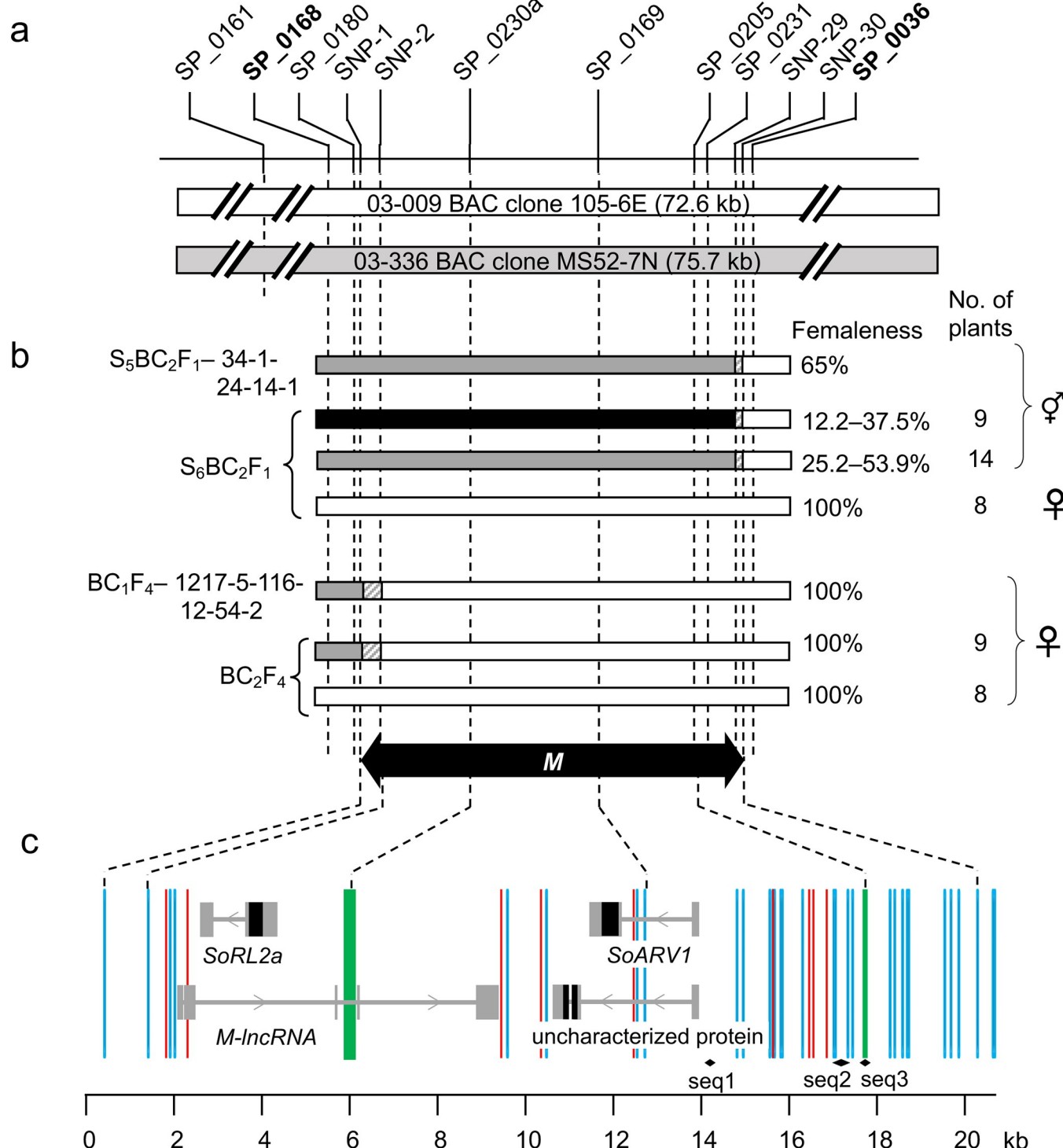

**Fig 2. Physical map of the *M* locus. a.** Schematic diagrams representing allelic bacterial artificial chromosome clones of the *M* locus from lines 03–009 and 03–336. Positions of the DNA markers are shown on the diagrams. **b.** Graphical genotypes and femaleness of progeny families from monoecious selections in $S_5BC_2F_1$ and $BC_1F_4$ populations produced by crosses between 03–009 and 03–336. Black boxes represent homozygous 03–336 segments, gray boxes indicate heterozygous 03–009/03–336 regions, and white boxes represent homozygous 03–009 segments. Indices of femaleness (expressed as the percentage of female flowers per plant) are shown to the right of the boxes. **c.** Schematic diagrams of protein-coding genes and a lncRNA locus predicted in the 19.5-kb *M* candidate region. Exons and coding regions are represented as gray and black boxes, respectively. Introns are depicted as horizontal gray lines. Gene directions are indicated by arrowheads. Vertical blue, red and green lines represent SNPs, small indels (≤ 7 bp) and structural variants (SV) found between lines 03–009 and 03–336. Diamonds (a–d) represent regions having significant sequence homology with the spinach *PISTILLATA* (*PI*) and *APETALA3* (*AP3*) genes. **seq1**, the

97-bp region, has homology with the third intron of *PI*; **seq2**, the 343-bp region, has homology with the 5th intron of *PI*, and the upstream region of *AP3*; **seq3**, the 75-bp region, has homology with the 6th intron of *AP3*.

were located in the 343-bp sequence ('**seq2**' in Fig 2C) with homology to *SpPI*, and the position of the 75-bp sequence ('**seq3**' in Fig 2C) with homology to *SpAP3* was matched with an SV (110-bp indel): the 75-bp sequence was present in the *M* locus of 03–009 but absent in that of 03–336. To make it easy to determine the genotypes at the *M* candidate region in subsequent analyses, a codominant marker, SP_0205, which targets the 110-bp indel, was developed (Fig 2 and S5 Table).

To identify the reproductive tissue–expressed gene(s) and the *M* gene from the *M*-candidate region, preliminary RNA-seq analysis of monoecious, female, and male plants was carried out without biological replicates. The female and male plants used in this analysis were from dioecious-line 03–009, and the monoecious plants were from line NIL-M, which is homozygous for *M* and produced by introgression of the *M* locus of line 03–336 into the genetic background of 03–009 (S3 Fig). Since 03–336 and 03–009 are late- and early-bolting/flowering lines, respectively, dioecious 03–009 and monoecious NIL-M lines with nearly identical bolting characteristics were used in this analysis. As shown in S8 Fig, *SoRL2a* was found to be expressed at relatively low levels in vegetative tissues (leaves) (FPKM = 0.6–4.1) and shoot apexes (FPKM = 1.6–14.6) from plants of all sexes. By contrast, its expression levels in the reproductive tissues (inflorescences) of the plants were relatively high and increased with maturation. It is also worth mentioning that *SoRL2a* was estimated to be expressed more in the inflorescences of the male (FPKM = 61.4–78.9) and monoecious (FPKM = 17.1–66.2) plants than in those of the female (FPKM = 5.8–29.7) plants. In plants of all sexes, the uncharacterized protein coding gene (g48913.t1 [Chr3_g019250.1]) had the highest TPM score (17.5–25.0) in shoot apexes under short-day (SD) conditions. *SoARV1* (g48912.t1 [Chr3_g019260.1]) was expressed at low levels in leaves (FPKM = 2.2–2.6), shoot apexes (FPKM = 2.0–5.9) and reproductive tissues (inflorescences) (FPKM = 2.0–3.3) of the male, monoecious, and female spinach plants. The levels of *M-lncRNA* expression were also estimated to be constitutively low (FPKM = 0.4–1.6) in plants of all sexes. There was no obvious trend associated with reproductive stages or sex types in the expression patterns of *SoARV1*, the uncharacterized protein-coding gene, or *M-lncRNA*. *SoRL2a* is a gene that encodes a small protein, which contains a SANT/MYB domain. *SoRL2a* is a homolog of the *RADIALIS* gene in Antirrhinum, which controls the asymmetry of flowers. A recent study revealed that a *RADIALIS* homolog, *DkRAD*, plays a crucial role in the reversion from male to hermaphrodite flowers in hexaploid persimmon by influencing gynoecium development. Since members of the *RADIALIS* family are

**Table 1. Genes located on the 19.5-kb candidate genomic region for the *M*-locus.**

| gene name | SOL_r1.1a gene id [16] | SOL_r2.0 gene id | blastp vs NR | E-value | blastp vs Araport11 | E-value |
|---|---|---|---|---|---|---|
| *SoRL2a* | g48915.t1 | Chr3_g019240.1 | XP_021847620.1 RADIALIS-like 2 [*S. oleracea*] | 4.60E-40 | AT1G75250.1 RAD-like 6 | 1.00E-36 |
| uncharacterized protein coding gene | g48913.t1 | Chr3_g019250.1 | XP_021847648.1 uncharacterized protein isoform X2 [*S. oleracea*] | 3.00E-46 | AT4G39235.2 hypothetical protein | 4.00E-33 |
| *SoARV1* | g48912.t1 | Chr3_g019260.1 | XP_021847636.1 protein arv1 homolog isoform X1 [*S. oleracea*] | 2.90E-118 | AT1G01020.1 ARV1 family protein | 5.00E-61 |
| | | | blastn vs NR | E-value | | |
| M-lncRNA | g48914.t2 | SOL_r2.0_LOC110787320 | XR_002533067.1 uncharacterized transcript variant X2, ncRNA [*S. oleracea*] | 0 | | |

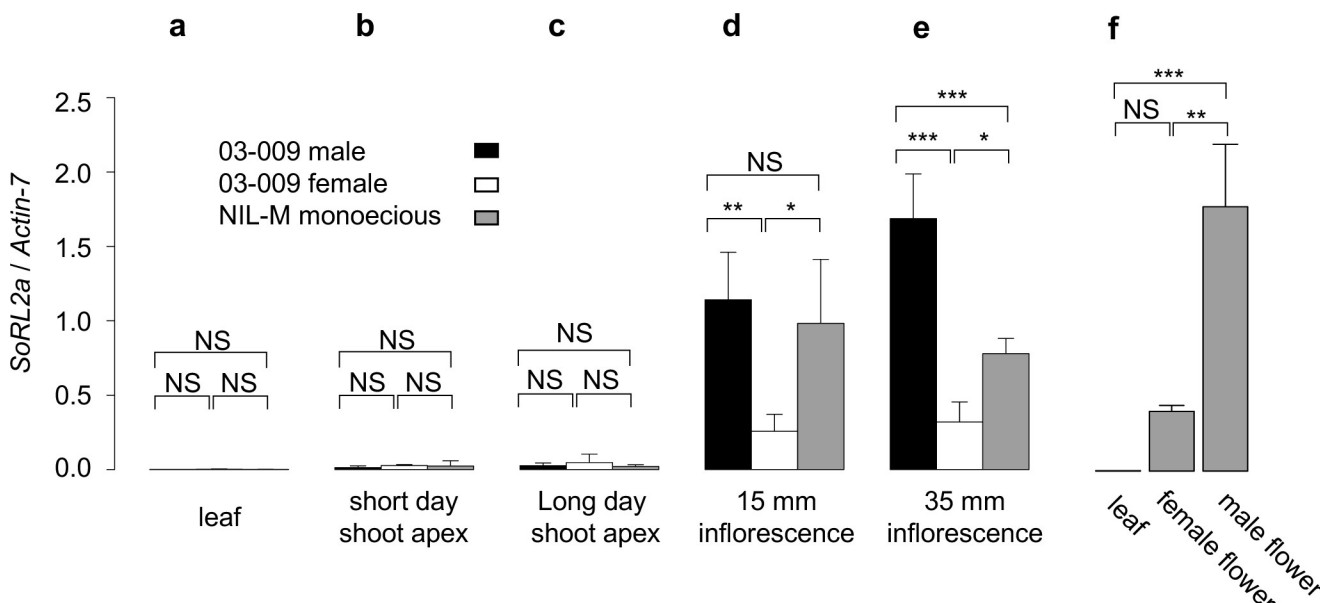

**Fig 3. RT-qPCR analysis of *SoRL2a*. a-e**, RT-qPCR analysis of *SoRL2a* in male and female plants from the dioecious line 03–009, and monoecious plants from line NIL-M; **f**, RT-qPCR analysis of *SoRL2a* in leaves, and female and male flowers of monoecious plants. The amount of *SoRL2a* mRNA was normalized to that of Actin-7.

involved in the formation of floral organs and sexual differentiation, and *SoRL2a* is preferentially expressed in reproductive organs, we considered that *SoRL2a* would be a promising candidate gene for male organogenesis and monoecious expression. We performed further verification to confirm this. It is worth noting that, among *RAD*-like genes in plants, *SoRL2a* was closely related to *RL2* (XM_010692578) in *Beta vulgaris* in our phylogenetic analysis (S9 Fig).

*SoRL2a* mRNA levels relative to *Actin-7* mRNA were quantified by RT-qPCR, to further characterize the *SoRL2a* expression profile in monoecious (NIL-M), female (03–009), and male (03–009) plants. As shown in Fig 3A–3E, the RT-qPCR assay reconfirmed that *SoRL2a* was predominantly expressed in inflorescences. Our RT-qPCR assay also showed that *SoRL2a* expression was 3.7–4.4-fold greater in the early-stage inflorescences of the male and monoecious plants than in those of the female plants. Furthermore, among inflorescences at the subsequent developmental stage, the male plant had the highest level of *SoRL2a* expression.

As shown in Fig 3F, RT-qPCR assay of tissues from inflorescences of monoecious plants showed that *SoRL2a* mRNA expression was approximately four-fold higher in male flowers than in female flowers. Its expression in leaves was shown to be almost absent, but not significantly different from that in female flowers. The lack of significance was likely due to the limited number of biological replicates (n = 3). Together with the results above, these findings indicate that *SoRL2a* mRNA is expressed in flowers of monoecious, female, and male plants, and the higher expression of *SoRL2a* mRNA in male and monoecious inflorescences may be due to its higher expression in male flowers than in female flowers.

To elucidate the detailed temporal and spatial expression profiles of *SoRL2a* mRNA within male and female flowers, *in situ* hybridization analysis was carried out on tissue sections from a male and a female inflorescence (S10 Fig). As shown in the photos of male flower sections (sections **a** and **a"** in S10 Fig), *SoRL2a* mRNA was expressed in the endothecia and middle layers of anthers before the start of meiosis (male flower 1 in sections **a** and **a"**) but was absent in

the anthers at the earlier stage (male flower 2 in sections **a** and **a"**). Furthermore, *SoRL2a* mRNA was found to persist for a short period after meiosis (sections **b** and **b"** in S10 Fig). *In situ* hybridization analysis of female flowers showed that *SoRL2a* mRNA was expressed at the base of the outer and inner integuments. However, *SoRL2a* mRNA was absent in the female flowers at the earlier stage when gynoecium primordia are initiated (data not shown). The higher expression of *SoRL2a* mRNA in male flowers than in female flowers was likely attributable to the regions where it was expressed in the male flowers being larger than those in the female flowers.

## Mapping of QTLs controlling femaleness in monoecious spinach

To investigate whether loci other than the *M* locus might be involved in the monoecious expression, female plants (*mm*) from line 03–009 (n = 7), monoecious plants (*MM*) from line 03–336 (n = 8), and $F_1$ (n = 8) and $F_2$ (n = 92) generations from the cross 03–009 x 03–336 were evaluated for femaleness. Monoecious plants from NIL-M (n = 8) were also subjected to the analysis.

As shown in S11 Fig, female plants of genotype *mm* from line 03–009 and monoecious plants of genotype *MM* from line 03–336, which were grown as control populations, displayed 100% and 4.8% femaleness scores on average, respectively. The $F_1$ plants (*Mm*) showed femaleness scores (64.7–77.4%, average = 72.7%) intermediate between those of the parental lines, which is consistent with our previous finding that *M* acts as an incomplete dominant gene [15]. Femaleness scores of $F_2$ plants from 03–009 x 03–336 ranged from 0% to 100% (average = 65.3%). When the $F_2$ plants were genotyped with marker SP_0205 at the *M* locus (S5 Table), 22 of the 25 *mm* plants (homozygous for the 03-009-derived allele of SP_0205) were found to show the female character (100% femaleness); the three exceptions were plants that were not strictly female but exhibited highly female monoecious conditions (92–97% femaleness). By using the 19 markers shown in S4 Table and S6 and S7 Figs, two (92% and 97% femaleness) of the three highly female monoecious plants were confirmed to be homozygous for the 03–009 genotype in the interval between SP_0008 and SP_0069 (~135.3 Mb in size), which includes the *M*-candidate region. The other one (93% femaleness) was also shown to be homozygous for the 03–009 genotype in the interval between SP_0125 and SP_0069 (~133.7 Mb in size), which includes the *M*-candidate region. Twenty-two *MM* plants (homozygous for the 03-336-derived allele of SP_0205) exhibited the monoecious character with relatively lower female scores (0–64%, average = 30.0%). Furthermore, 48 of 49 *Mm* plants (homozygous and heterozygous for the 03-336-derived allele of SP_0205) exhibited monoecious conditions with a wide range of femaleness scores from 0% to 99% (average = 65.2%), with the exception being a single plant with a strict female condition with a femaleness score of 100%. This result was likely attributable to the incomplete dominant nature of the *M* gene: the *Mm* genotype often produces plants with higher femaleness scores than the *MM*. These results supported our conclusion that *M* is the major factor regulating the monoecious expression [15, 17].

The respective *MM* and *Mm* plants in the $F_2$ population had a wider range of femaleness scores compared to monoecious plants from 03–336 (*MM*) and the $F_1$ generation (*Mm*) (S11 Fig). Furthermore, monoecious plants from NIL-M (n = 8) that were homozygous for *M* and were produced by introgression of the *M* locus of 03–336 into the genetic background of 03–009 showed femaleness scores (38–88%, average = 63.5%) significantly higher than those exhibited by the 03–336 plants (Exact Wilcoxon rank sum test, $W = 64$, $p < 0.001$). These findings may suggest that monoecious expression (degree of femaleness) is controlled by not only the *M* locus but also other loci.

In order to identify the additional loci for monoecism, other than *M*, QTL analysis for femaleness was performed using the 03–336 x 03–009 F2 population. We detected three QTLs with LOD scores greater than 4.0; these were designated *qFem2.1* (LOD = 8.15), *qFem3.1* (LOD = 21.8), and *qFem6.1* (LOD = 4.0) on LGs 2, 3, and 6, respectively (Fig 1). However, after performing the permutation test, only two (*qFem2.1* and *qFem3.1*) QTLs remained significant. Referring to our pseudomolecules, the chromosomal region for *qFem3.1* (99% CI) was found to harbor the *M* candidate region.

## *qFem2.1* and *qFem6.1* act as modifiers of the *M* gene function

Next, we plotted the means of femaleness scores in the $F_2$ population against genotypes at the QTLs, as shown in S12 Fig. The plots suggested that *qFem2.1* and *qFem6.1* may affect femaleness in the monoecious plants carrying the *M* allele—i.e., *MM* and *Mm* plants in the $F_2$ population tended to exhibit lower femaleness as the dose of 03-336-derived alleles for *qFem2.1* and *qFem6.1* increased. Still, this trend was not observed in *mm* plants. The "fitqtl" function of the software package R/QTL detected the interaction effect between *qFem3.1/M* and *qFem2.1* (ANOVA *F*-test *p*-value < 0.05; S9 Table) in the $F_2$ population but failed to detect an interaction effect between *qFem3.1/M* and *qFem6.1* (data not shown). The plants used in the analyses described above were grown at a high temperature (25˚C) during a reproductive developmental stage (the periods when plants were grown under long-day photoperiods), since the high-temperature conditions bring about a shift towards maleness in monoecious plants [3, 43]. Therefore, compared to low-temperature conditions, high-temperature conditions can be expected to make *Mm* plants exhibit obvious monoecious conditions with relatively low femaleness scores. To verify the effect of *qFem2.1* and *qFem6.1* on femaleness, the $F_2$ progeny plants from the cross between 03–336 x 03–009 were grown again, and then 72 *MM* plants were selected from the $F_2$ progeny. The *MM* $F_2$ plants and control plants (females [*mm*] from 03–009, monoecious [*MM*] from NIL-M and 03–336) were grown under a low temperature (20˚C), since these *MM* plants were expected to exhibit a wide range of femaleness scores from low to high depending on their genetic backgrounds. Subsequently, the *MM* plants were genotyped for *qFem2.1* and *qFem6.1* using dCAPS markers (SP_0319 and SP_0325; S5 Table).

As shown in Fig 4A, female plants from 03–009 displayed the strictly female condition (100% femaleness score), and femaleness scores exhibited by the plants from NIL-M (n = 11) and 03–336 (n = 3) were 94.9% and 5.1% on average, respectively: the difference in femaleness scores of the two monoecious lines grown under the low-temperature condition was more remarkable than that under the high-temperature condition (S11 Fig). In the 72 *MM* plants selected from the $F_2$ generation, the mean of femaleness scores exhibited by plants homozygous for 03-336-derived alleles for *qFem2.1* and *qFem6.1* was significantly lower than that of plants heterozygous and homozygous for 03-009-derived alleles (Fig 4A). Furthermore, the effect of *qFem2.1* on femaleness was confirmed again in the $F_3$ progeny (n = 37) from an $F_2$ plant with the *MM* genotype, which was heterozygous at the *qFem2.1* locus, but homozygous for 03-009-derived alleles at the *qFem6.1* locus (Fig 4B). The examination for *qFem6.1* using the $F_3$ generation was not carried out due to the failure to produce $F_3$ progeny from $F_2$ plants heterozygous at the *qFem6.1* locus.

## Transcriptome analysis of 03–009, 03–336 and NIL-M

To identify candidates for genes involved in the regulation of femaleness in monoecious plants, transcriptome profiles of early-stage inflorescences from highly male monoecious (03–336), highly female monoecious (NIL-M), and female (03–009) plants grown at a low temperature (20˚C) were analyzed by RNA-seq. As shown in S13 Fig, 1,834 up-regulated and 2,153

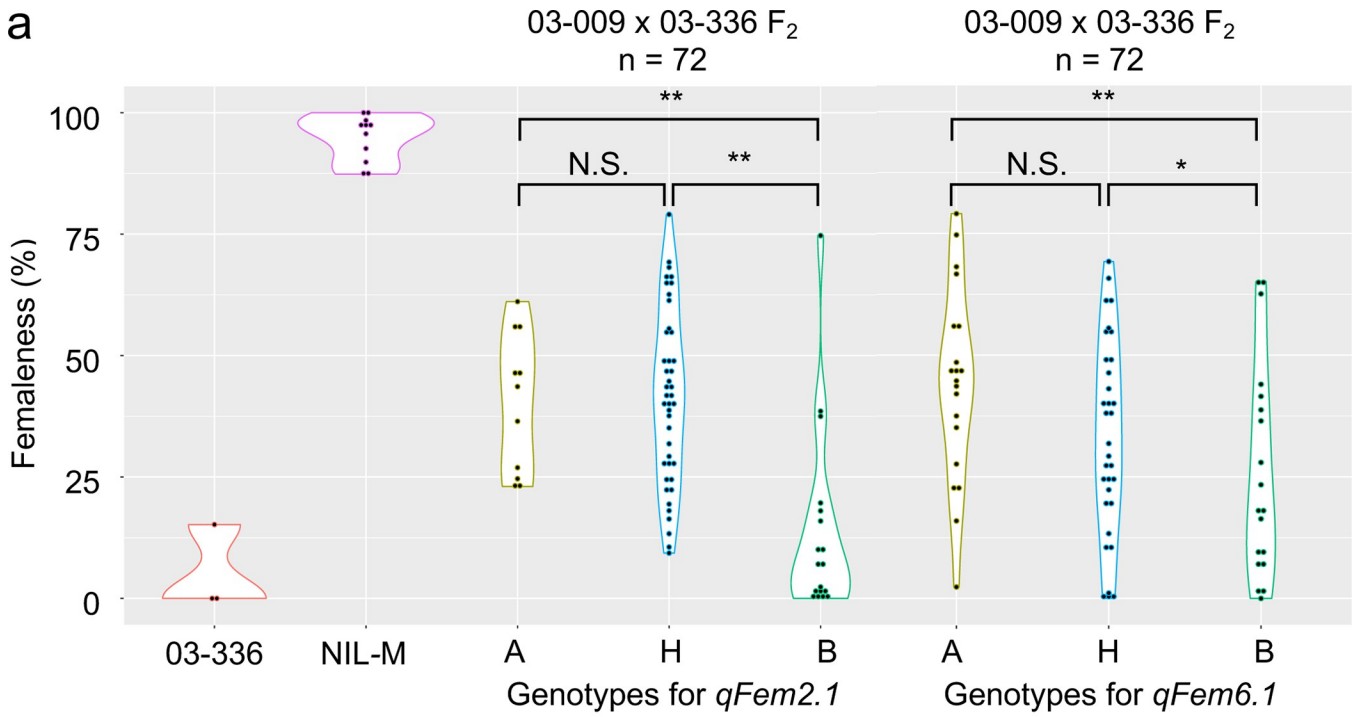

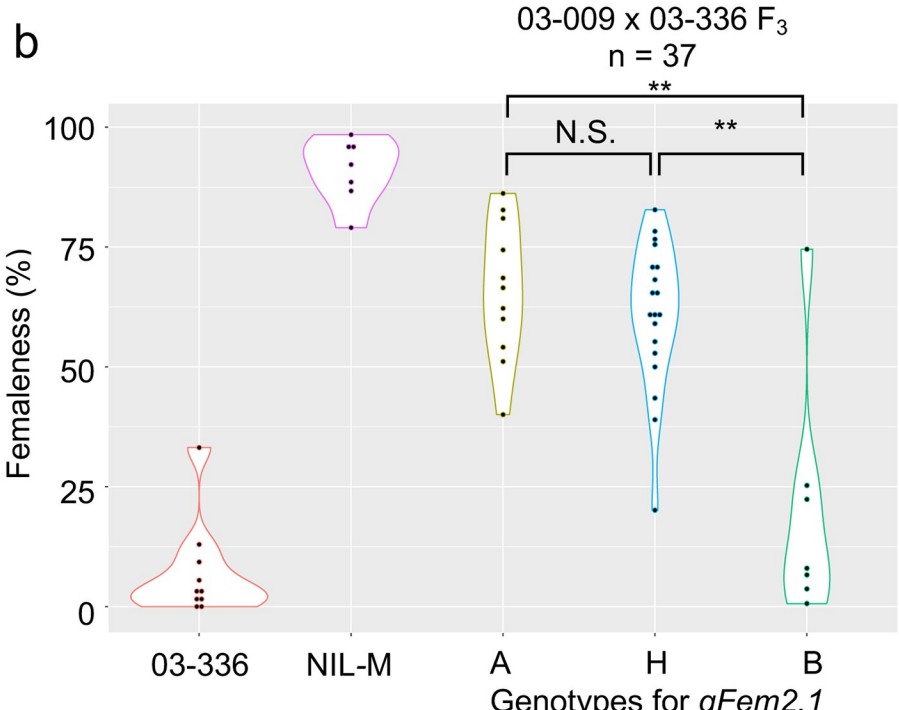

**Fig 4. Violin plots showing the distribution of femaleness exhibited by *MM* plants selected from F$_2$ and F$_3$ progeny of the cross between 03–009 and 03–336.** Statistical significance was determined by Tukey-Kramer test (N.S. $p \geq 0.05$; * $p < 0.05$; ** $p < 0.01$). A, homozygous for the 03–009 allele; H, heterozygous for the 03–009 and 03–336 alleles; B, homozygous for the 03–336 allele.

down-regulated differentially expressed genes (DEGs) were identified by comparing 03–336 vs. 03–009, NIL-M vs. 03–009, and 03–336 vs. NIL-M plants. However, the protein-coding genes in the *M*-candidate region were not included in the DEGs.

One hundred thirty-six genes were found to be commonly up-regulated in highly male monoecious (03–336) and highly female monoecious (NIL-M) plants compared with their expression in female plants (03–009). These genes were further characterized by GO enrichment analysis (S13 Fig). As shown in S10 Table, the shared up-regulated DEGs were significantly enriched with GO terms including 'stamen formation' (GO: 0048455) and 'specification of floral organ identity' (GO: 0010093). Both DEGs (g3323.t1 [Chr5_g058930.1] and g3324.t1 [Chr5_g058920.1]) annotated with the GO terms were homologs (*SpPI*) for Arabidopsis *PISTILLATA* (*PI*). However, the homologs were located on Chr5 (LG5), which does not harbor an *M* locus or other QTLs. Karrikin (GO:0080167), RNA 5'-end processing (GO:0000966), behavior (GO:0007610), and vernalization response (GO:0010048) were also highly ranked among the enriched, shared up-regulated DEGs. 175 down-regulated DEGs were found to be shared between the comparison pairs 03–336 vs. 03–009 and NIL-M vs. 03–009 (S13 Fig). However, the shared DEGs were significantly enriched with biological process GO terms that were not directly related to flower organ formation or reproductive development processes, such as mismatch repair (GO:0006298), wax biosynthetic process (GO:0010025), and mitochondrial mRNA processing (GO:0090615) (S11 Table).

Eight-hundred thirty-six genes were found to be up-regulated in highly male monoecious (03–336) plants compared to both highly female monoecious (NIL-M) and female (03–009) plants and were characterized based on GO enrichment analysis (S13 Fig). The up-regulated DEGs also included *PI* homologs and were shown to be significantly enriched with four GO terms related to reproductive organogenesis (GO:0009554, GO:0048658, GO:0009556, and GO:0048455) (S12 Table). However, the DEGs were most significantly enriched with the GO term 'defense response to fungus' (GO:0050832). 'Auxin export across the plasma membrane' (GO:0010315) was another GO term significantly enriched among the DEGs.

One thousand and five genes were found to be down-regulated in highly male monoecious (03–336) plants compared to both highly female monoecious (NIL-M) and female (03–009) plants (S13 Fig). The down-regulated DEGs were shown to be significantly enriched with GO terms related to the reproductive process, such as 'vernalization response' (GO:0010048), 'regulation of flower development' (GO:0009909) and 'positive regulation of reproductive process' (GO:2000243) (S13 Table). The down-regulated DEGs were also significantly enriched with genes related to defense response (GO:0006952 and GO:0098542), as in the up-regulated DEGs.

## Candidate genes for the QTLs *qFem2.1* and *qFem6.1*

Among the following conditions, the genes satisfying conditions (1) & (2) or conditions (1) & (3) were selected from the *qFem2.1* region (3.2 Mb, 95%CI) as the candidate genes. (1) The genes were expressed above 1 TPM, in early-stage inflorescences of at least one line. (2) The genes were predicted to be associated with reproductive, floral-organogenesis, or hormone metabolism/transport/response, and had nucleotide variants classified as exerting 'HIGH' or 'MODERATE' impact that were assumed to disrupt functions or change the effectiveness of encoded proteins. (3) The genes were predicted to be associated with the biological processes mentioned above, to be differentially expressed among the different lines (03–009, NIL-M and 03–336), and to have nucleotide variants classified as exerting 'LOW' or 'MODIFIER' impact that were assumed to be unlikely to change the functions of encoded proteins.

Four candidate genes were identified after searching the *qFem2.1* region (S14 Table). The candidate genes g23600.t1 (Chr2_g000510.1) and g23706.t3 (Chr2_g001540.1), which had

HIGH- and MODERATE-impact variants, are homologs of the Arabidopsis genes AT5G50850 (*MAB1*) and AT5G13010 (*EMB3011 / PINP1*) associated with auxin transport and auxin-mediated organ development, respectively [44, 45]. The gene g23617.t1 (Chr2_g000670.1) with a MODIFIER-impact variant was predicted to encode an auxin-responsive protein and was expressed more highly in 03–336 than in NIL-M or 03–009. The gene g23612.t1 (Chr2_g000620.1) had a MODIFIER-impact variant and is a homolog of the Arabidopsis gene (AT5G14070.1.1/*ROXY2*) encoding a thioredoxin superfamily protein that was shown to be involved in anther development [46] and to be expressed more highly in 03–336 than in NIL-M or 03–009.

The two genes satisfying only conditions (1) & (2) above were selected from the *qFem6.1* region (57.1 Mb, 95%CI) as the candidate genes, since this region was too wide to select candidates based on the same conditions as in the case of *qFem2.1* (S14 Table). G49564.t1 (Chr6_g038890.1) and g13001.t2 (Chr6_g032140.1), which had MODERATE-impact variants, were predicted to encode LRR receptor-like kinase (LRR-RLK). The former gene is a homolog of the Arabidopsis gene, *CLAVATA3 INSENSITIVE RECEPTOR KINASEs* (*CIKs* / AT3G02130), which was shown to regulate early anther development [47]. The latter one is a homolog of the Arabidopsis gene and *CLAVATA 1* (*CLV1* / AT1G75820), respectively, which are required for maintaining floral meristem identity [48].

## Discussion

In this study, we successfully constructed a set of chromosome-scale pseudomolecules, SOL_r2.0_pseudomolecule, representing the genome of a dioecious line 03–009, by assembling PacBio long reads with the help of Hi-C-based chromatin proximity information and a RAD-Seq-derived SNP linkage map.

Our previous version of the genome assembly, SOL_r1.1 [20], covered over 90% of the whole genome and had a high degree of continuity (N50 = 11.3 Mb, LAI = 15.91). However, we discovered that 29 out of the 112 scaffolds in SOL_r1.1, which were anchored with the linkage map, were chimeric due to scaffolding errors. This may have been due to the size of the spinach genome, which is relatively large (~1000 Mb) and contains many repetitive elements. Additionally, the mate-pair libraries used to construct SOL_r1.1 were limited to 3, 6, 10, and 15 kb inserts, which provided proximity information for only a small area. Hi-C analysis generates genome-wide chromatin interaction data, providing genomic proximity information with a scale ranging from 1 kb to 1 Mb or longer, to assign, order, and orient genomic sequences, achieving high accuracy in genome assembly for humans and goats, both of which have large-size genomes [25, 49]. Therefore, combining long-read sequencing technologies with Hi-C proximity data may be appropriate to construct chromosome-scale assembly for this genome, rather than combining those with mate-pair reads. Finally, SOL_r2.0_pseudomolecule had a total length of 879.2 Mb and scored 97.0 for BUSCO (gene completeness) and 19.94 for LAI (sequence continuity). It outperformed the SOL_r1.0_pseudomolecule constructed from the genome assembly SOL_r1.1 in terms of total length, BUSCO score (81.5), and LAI score (17.08) (S1 Table). In recent years, by combining long-read and Hi-C data, pseudomolecules of the genomes for the spinach lines Monoe-Viroflay, Viroflay, and Cornell-No. 9 have been constructed [14, 50] (S1 Table). The pseudomolecules for Monoe-Viroflay and Virelay have good gene completeness scores (BUSCO completeness ≥ 97%) and high assembly continuity scores (LAI > 19.0). Similarly, the SOL_r2.0_pseudomolecule is of high quality and may be useful as a reference genome for future genetic studies toward identifying loci that control agronomic traits. Monoe-Viroflay and Viroflay likely derive from a European heirloom cultivar, since 'Viroflay' is generally known to be an old French spinach variety.

'Monoe-Viroflay' was reported to be provided by a European company (Enza Zaden, https://www.enzazaden.com/). In contrast, the breeding line 03–009 originates from a Japanese cultivar. Although the details of its pedigree information are not publicly available, line 03–009 is known to exhibit typical phenotypes observed in East Asian cultivars: early bolting, hastate and flat leaves, and production of prickly fruits [20]. Given that the use of only a single reference could bias the analysis toward a comparison within the chromosomal regions highly conserved with the reference due to difficulties in adequately aligning short-reads to highly divergent regions, the pseudomolecules for the 03–009 genome, SOL_r2.0_pseudomolecule, might be helpful as a reference for East Asian spinach genomes [51]. In particular, SOL_r2.0_pseudomolecule can be expected to serve as a reference to obtain highly accurate variant information in spinach germplasm collections derived from native varieties in East Asia, including Japan. Furthermore, the variant information obtained here will further understanding of the genetic structure of these genetic resource collections and is expected to be useful when using these genetic resource collections as breeding materials.

This study successfully identified three QTLs regulating femaleness in monoecious spinach plants by using variant data of $F_2$ progeny plants from the cross between 03–009 (female) and 03–336 (highly male monoecious) detected from SOL_r2.0_pseudomolecule. These QTLs were mapped to the SOL_r2.0_pseudomolecule, and the QTL with the highest LOD score, *qFem3.1*, corresponds to the major locus (*M*) for monoecism we previously identified [15–17]. PCR-based DNA markers targeting the QTLs were also successfully designed by referring to variants detected in the SOL_r2.0_pseudomolecule. These results indicate that the SOL_r2.0_pseudomolecule is suitable for NGS-based variant detection and genetic mapping studies and is a platform with enormous potential for developing PCR-based DNA markers that can aid in agronomic trait selection. Furthermore, our research provides valuable insights into the complex genetic regulatory mechanisms that govern the variability of femaleness in monoecious spinach plants. We have elucidated interaction patterns of the three QTLs: the *qFem3.1* (*M* locus) QTL is a crucial factor that is necessary for monoecious expression, while the *qFem2.1* and *qFem6.1* QTLs serve as modifiers for the *M* locus that affect femaleness. These findings will be valuable for spinach hybrid breeding programs using parental lines with highly male and highly female monoecious characters. The monoecious line 03–336 used in our study is a useful source of highly male monoecious characters that are desirable for paternal parents for hybrids. By pyramiding *qFem2.1*, *qFem6.1*, and *M* loci derived from monoecious line 03–336, using the variants and markers at the loci, it will be possible to efficiently produce paternal breeding lines. Similarly, the monoecious line NIL-M can be used as a source of highly female monoecious characters by utilizing the variants and markers present in the loci.

Of the three protein-coding genes in the *M* candidate region (the 19.5-kb interval between SNP1 and SNP30; Fig 2), only *SoRL2a* was found to be expressed in reproductive organs. *SoRL2a* is a homolog of *RADIALIS* that has been well characterized as a gene encoding a SANT/MYB domain transcription factor controlling floral asymmetry in *Antirrhinum majus* [52]. In tomato, a *RADIALIS*-like gene, *FSM1*, controls cell expansion and fruit development [53]. In Arabidopsis, the *RADIALIS-like 2* (*RSM1*) gene was reported to be involved in early morphogenesis [54, 55]. A recent study reported that a *RADIALIS* homolog, *DkRAD*, is a key factor for reversions from male to hermaphrodite flowers via gynoecium development in the monoecious hexaploid persimmon [56]. *DkRAD* was shown to be induced by abscisic acid and the cytokinin signaling responsible for the reversions to hermaphrodite flowers. However, in dioecious diploid persimmon, abscisic acid and cytokinin treatments cannot induce the reversions to hermaphrodite flowers, and thus cannot induce *DkRAD* expression. By contrast, *SoRL2a* was shown to be expressed not only in monoecious plants but also male and female

plants, suggesting that the role of this gene may be different from that of *DkRAD in* monoecious persimmon.

Sex differentiation in spinach flowers is not involved in the developmental arrest or abortion of one sex organ in bisexual floral primordia, since rudimentary opposite-sex organs were not found in unisexual flowers of spinach [57]. The floral sex differentiation system in spinach is, therefore, assumed to regulate the initiation of the stamen and pistil primordia. Spinach B class genes (*SoPI* and *SoAP3*) were shown to be involved in sex differentiation, since the suppression of the genes in male plants resulted in homeotic transformation of stamens into carpels and the presence of a fourth whorl [58]. In our study, *SoPI* was observed to be up-regulated in monoecious plants compared to female plants. However, the expression of *SoRL2a* mRNA was not observed in floral organ primordia, though it was observed in developing anthers and pistils (S10 Fig). Therefore, we cannot reasonably conclude that *SoRL2a* directly regulates the initiation of the sex organs in cooperation with the floral organ identity genes. According to Takahata et al. [16], the region of the spinach chromosome that contains the *M* locus shares a high degree of gene synteny with the corresponding region of the hermaphroditic species *Beta vulgaris* (sugar beet). Sugar beet belongs to the *Amaranthaceae* family, which also includes spinach. The chromosomal region in sugar beet that corresponds to the spinach *M* locus contains *BvRL2* (XM_010692578), the ortholog of *SoRL2a* (S9 Fig). Since the function of *BvRL2* might be similar to that of *SoRL2a* in the hermaphroditic species, it is reasonable to assume that *SoRL2a* does not play a role in regulating floral sex determination, but could be involved in the development of anthers and ovules based on its expression profile.

Papaya is a species that has three types of individuals: male, female, and hermaphrodite. Profiling analyses of small non-coding RNAs (sRNA) from the flowers of papaya showed that in male flowers, most of the expressed miRNAs were related to the auxin signaling pathways. On the other hand, the miRNAs that are more highly expressed in female flowers have the potential to regulate the apical meristem identity genes. These findings suggest that these miRNAs could play critical roles in the sex expression in papaya [59]. In diploid persimmon, siRNAs generated from a Y-chromosomal gene, designated *OGI*, play a role in male determination by suppressing the expression of its autosomal paralog, *MeGI*, which is a male function suppressor [60]. Furthermore, epigenetic regulation, triggered by *OGI*-derived siRNA, of *MeGI* was shown to be responsible for monoecious expression in hexaploid persimmon [61]. A recent study found that two Y-chromosomal genes are involved in poplar sex determination: one produces siRNAs that repress the expression of its autosomal paralog, which acts as a female-promoting factor, while the other generates long non-coding RNAs promoting maleness [62]. In this context, we should consider the possibility that unidentified non-coding RNA might be involved in the monoecious expression in spinach. Given that the *SoPI* and *SoAP3*-homologous sequences in the *M* candidate region are transcribed and generate siRNA, the *M* locus could control the monoecious expression via modulation of the B class genes. Nevertheless, more empirical evidence is necessary to verify this speculation.

The shared up-regulated DEGs were shown to be enriched with the GO term 'stamen formation' (GO: 0048455), as expected (S10 and S12 Tables). The up-regulated DEGs with the GO term include two copies of *PI* homologs, suggesting that the B class genes play a crucial role in the monoecious expression. Chailakhyan and Khryanin [63] reported that the exogenous application of gibberellic acid (GA3) on dioecious spinach plants resulted in an increase in the proportion of male plants. The level of expression of the spinach homolog of Arabidopsis *GAI* (AT1G14920), a member of the GA-repressor DELLA family, was reported to be significantly lower in males than in females [64]. Subsequent knockdown analysis of the spinach *GAI* suggested that it regulates the B class genes and is involved in sex determination in

spinach [64]. However, our transcriptome analysis did not detect significant differences in the expression levels of spinach homologs (g15409.t1 [Chr5_g029780.1] and g20972.t1 [Chr1_g022500.1]) of Arabidopsis DELLA family members (*GAI* and *RGA1* [AT2G01570]) between the female and monoecious plants (data not shown), and our GO enrichment analysis failed to reveal an association of GA with the monoecious expression. Furthermore, the spinach *GAI* and *RGA1* homologs are located on Chr.1 and Chr.5, which are unrelated to the QTLs detected herein, suggesting that the regulatory mechanisms of the B class genes might be different between male and monoecious plants.

The up-regulated and down-regulated DEGs were found to be enriched with GO terms including 'bacterium response' (GO:0009617) and 'defense response' (GO:0006952, GO:0050832, GO:0098542) (S11 and S12 Tables). This might suggest that pathways for sex-differentiation and stress/defense response have some commonalities. Indeed, stress-response-related genes (e.g., defense response, drought/oxidative stress response) were shown to be involved in the sex-differentiation of persimmon [56]. We note that the RNA helicase gene (g23706.t3 [Chr2_g001540.1]), which was located in the *qFem2.1* region, was annotated with GO:0006952 (defense response) and was included among down-regulated DEGs shared between 03–336 vs. 03–009 and 03–336 vs. NIL-M (S13 Table). Its Arabidopsis homolog (AT5G13010/ EMB3011 /*PINP1*) was previously shown to be associated not only with auxin-mediated organ development but also with plant immunity [44, 65].

The genes annotated with the GO term 'cellular response to toxic substance' (GO:0097237) in the up-regulated DEGs (S10 Table) encode a homolog of Arabidopsis UDP-glucosyltrans-ferase (AT1G05680) induced by $H_2O_2$, and peroxidase superfamily proteins. Interestingly, UDP-glucosyltransferase was shown to be involved in auxin homeostasis [66]. Furthermore, the genes annotated with the GO term 'auxin export across the plasma membrane' (GO:0010315) were enriched among up-regulated DEGs shared between the comparison pairs 03–336 vs. 03–009 and 03–336 vs. NIL-M (S12 Table). Among genes located in the *qfem2.1* and *qfem6.1* regions that carry variants potentially affecting their function, auxin-related genes were found in addition to floral-organogenesis genes.

Auxins are involved in various plant developmental processes, including floral meristem initiation, floral organ boundary establishment, floral organ outgrowth and floral meristem determinacy [67]. *ARF6* and *ARF8*, members of the auxin response factor gene family, were reported to control stamen elongation and flower maturation in Arabidopsis [68]. A double mutant with defects of both genes has short petals, short stamen filaments, late dehiscent anthers, and immature gynoecia. *AUXIN RESPONSE FACTOR 3* is involved in auxin-regulated gynecium patterning [69]. The Arabidopsis SUPERMAN (SUP) gene encodes a C2H2 zinc finger transcription factor that is expressed at the boundary of the stamen and carpel whorls [70]. This gene negatively regulates the expression of the auxin biosynthesis genes *YUCCA1/4* [71]. In sup mutants, the auxin levels increase at the boundary, leading to an increase in the number and the prolonged maintenance of floral stem cells [70, 71]. This results in supernumerary stamens and reduced or absent carpel tissue, as the fate of cells in whorl 4 changes from female to male [67]. Auxin plays a crucial role in reproductive organogenesis, which is why it may be involved in the sexual differentiation of flowers in certain dioecious and monoecious plants. In fact, auxins were reported to be potentially involved in sex differentiation in several dioecious plants, including spinach. Moreover, auxin application was shown to raise the proportion of female plants in a dioecious spinach cultivar with an even ratio of female to male plants under control conditions [67]. Comparative transcriptomic analysis of male and female floral buds in papaya suggested that auxin transporters might be involved in sex differentiation [72]. Heslop-Harrison [73] provided evidence that treating male dioecious hemp with auxins resulted in the development of female flowers. A SAUR (small auxin-up

RNA) gene was reported to be one of the putative candidate genes for controlling sexual traits in grapevine [74]. All these facts suggest that the auxin-related genes in the *qfem2.1* region could be promising candidate genes for mediating the sexual differentiation of flowers and regulating femaleness in monoecious spinach.

*CLAVATA3* (*CLV3*) encodes a small secreted peptide that negatively regulates stem cell activity in Arabidopsis [75]. A recent study suggested that *CLV3* plays a role in regulating the size of floral meristems in coordination with *AGAMOUS* (*AG*) and *SUPERMAN* (*SUP*) [76]. Proper regulation of floral meristem activity is crucial for the normal development of reproductive organs, as suggested by *sup* mutants. Therefore, it is possible that homologs of *CLV3* could affect the sexual differentiation of flowers. A *CLV3*-like gene was recently identified as a potential sex-determining factor (gynoecium suppression factor) in *Silene latifolia* [77]. In the context of monoecious spinach, variants in the homologs of Arabidopsis *CLAVATA1* and *RPK2* [78, 79], which are receptors of *CLV3*, located in the *qfem6.1* region, may also be considered to affect femaleness. However, these ideas are preliminary and require further investigation.

## Supporting information

**S1 Fig. Crossing scheme to produce self–pollinated progeny families from a monoecious selection in a 03–009×03–336 BC$_2$F$_1$ population.**
(PDF)

**S2 Fig. Crossing scheme to produce 03–009×03–336 BC$_1$F$_4$.**
(PDF)

**S3 Fig. Crossing scheme to obtain the line NIL–M bearing the *XXMM* genotype in the background of the dioecious line 03–009.**
(PDF)

**S4 Fig. Representative images of plant tissues used for RNA extraction.**
(PDF)

**S5 Fig. Hi-C contact map of the chromosomal pseudomolecules SOL_r2.0_pseudomolecule.**
(PDF)

**S6 Fig. Schematic diagrams showing the physical position of a chromosomal region harboring the *qFem3.1 and M* loci. a.** A schematic diagram showing the physical positions of *qFem3.1* and DNA markers on a pseudomolecule, Chr3. **b.** Graphical genotypes and femaleness of self–pollinated progeny families from a monoecious selection in an S$_2$BC$_2$F$_1$ population produced by crosses between 03–009 and 03–336. Black boxes represent homozygous 03–336 segments, gray boxes indicate heterozygous 03–009/03–336 regions, and white boxes represent homozygous 03–009 segments. Marker positions and names are indicated above the boxes. Indices of femaleness (expressed as the percentage of female flowers per plant) are shown to the right of the boxes.
(PDF)

**S7 Fig. Graphical genotypes and femaleness of progeny families from monoecious selections in a S$_3$BC$_2$F$_1$ and a F$_2$ population produced by crosses between 03–009 and 03–336.** Black boxes represent homozygous 03–336 segments, gray boxes indicate heterozygous 03–009/03–336 regions, and white boxes represent homozygous 03–009 segments. Markers are indicated above the boxes. Indices of femaleness (expressed as the percentage of female flowers

per plant) are shown to the right of the boxes.
(PDF)

**S8 Fig. Expression levels of the genes located in the 19.5-kb chromosomal region candidate for the monoecious locus measured by RNA-seq.**
(PDF)

**S9 Fig. Maximum likelihood tree of RADIALS-like genes.** *S. oleracea* (*So*), *Beta vulgaris* (*Bv*), *Arabidopsis thaliana* (*At*), *Diospyros kaki* (*Dk*), *Antirrhinum majus* (*Am*) and *Oryza sativa* (*Os*). Bootstrap values above 50% from 1,000 iterations are indicated below the nodes.
(PDF)

**S10 Fig. Spatial expression analysis of *SoRL2a*. a**-**c**, Hematoxylin-eosin-stained sections of flower(s) from a male (**a** and **b**) and a female (**c**) plant. Two male flowers at different developmental stages are shown in panel **a** and numbered 1 and 2; **a'**-**c'**, *In situ* hybridization of flower sections from a male (**a'** and **b'**) and a female (**c'**) plant with anti-sense *SoRL2a* riboprobes; **a''**-**c''**, Flower sections from a male (**a''**) and a female (**b''** and **c''**) plant probed with sense *SoRL2a*. E, epidermis; En, endothecium; ML, middle layer; MMC, microspore mother cells; T, tapetum; Tds, tetrads; G, gynoecium; II, inner integument; OI, outer integument; Nu, nucellus; Ch, chalaza.
(PDF)

**S11 Fig. Frequency distribution of the femaleness in the 96 $F_2$ progeny plants from the cross between 03–009 and 03–336.** Arrowheads and horizontal bars indicate average numbers and ranges, respectively, of femaleness of the parental lines, 03–009 and 03–336, the $F_1$ progeny, and NIL-M ($S_2BC_5F_1$).
(PDF)

**S12 Fig. Means of femaleness scores in the 03–009 x 03–336 $F_2$ population against genotypes at the QTLs.** A, homozygous for the 03-009-derived allele; H, heterozygous for the 03–009 and 03-336-derived alleles; B, homozygous for the 03-336-derived allele.
(PDF)

**S13 Fig. Venn diagrams showing the genes differentially expressed between early-stage inflorescences from the spinach lines 03–336, NIL-M and 03–009.** 03–336, highly male monoecious; NIL-M, highly female monoecious; 03–009, female.
(PDF)

**S1 Table. Status of genome assemblies for spinach line 03–009 in the five assembly steps.**
(PDF)

**S2 Table. Hi-C mate-pair reads used in this study.**
(PDF)

**S3 Table. Summary of whole genome sequencing reads used for variant discovery.**
(PDF)

**S4 Table. Primer sequences and annealing temperatures of the spinach genetic markers used for fine mapping of the M-locus.**
(PDF)

**S5 Table. Primer sequences and annealing temperatures of the spinach genetic markers flanking the QTLs controlling monoecious expression.**
(PDF)

**S6 Table. Primers used for the RT-qPCR assay.**
(PDF)

**S7 Table. RNA-Seq reads used for transcriptome analysis.**
(PDF)

**S8 Table. A linkage map and pseudomolecules of spinach.**
(PDF)

**S9 Table. ANOVA table showing the genetic interaction between loci *qFem3.1/M* and *qFem2.1*.**
(PDF)

**S10 Table. BP GO terms enriched in 136 up-regulated DEGs shared between the comparison pairs 03–336 vs. 03–009 and NIL-M vs. 03–009.**
(PDF)

**S11 Table. BP GO terms enriched in 175 down-regulated-DEGs shared between the comparison pairs 03–336 vs. 03–009 and NIL-M vs. 03–009.**
(PDF)

**S12 Table. BP GO terms enriched in 836 up-regulated DEGs shared between the comparison pairs 03–336 vs. 03–009 and 03–336 vs. NIL-M.**
(PDF)

**S13 Table. BP GO terms enriched in 1005 down-regulated DEGs shared between the comparison pairs 03–336 vs. 03–009 and 03–336 vs. NIL-M.**
(PDF)

**S14 Table. Candidate genes underlying QTLs for the monoecious condition.**
(PDF)

## Acknowledgments

Computations were partially performed on the NIG supercomputer at the ROIS National Institute of Genetics. The authors thank the Tohoku Seed Co. (Utsunomiya, Tochigi, Japan) for providing the spinach breeding lines used in this study. We also appreciate the technical assistance provided by Mrs. H. Yokomoto.

## Author Contributions

**Conceptualization:** Hideki Hirakawa, Yasuyuki Onodera.

**Data curation:** Babil Pachakkil, Keisuke Tanaka, Yasuyuki Onodera.

**Formal analysis:** Kaoru Yamano, Akane Haseda, Keisuke Iwabuchi, Takayuki Osabe, Yuki Sudo, Babil Pachakkil, Keisuke Tanaka, Hideki Hirakawa, Yasuyuki Onodera.

**Funding acquisition:** Hideki Hirakawa, Yasuyuki Onodera.

**Investigation:** Kaoru Yamano, Akane Haseda, Keisuke Iwabuchi, Takayuki Osabe, Yuki Sudo, Yutaka Suzuki, Atsushi Toyoda, Hideki Hirakawa, Yasuyuki Onodera.

**Methodology:** Yasuyuki Onodera.

**Project administration:** Yasuyuki Onodera.

**Supervision:** Yasuyuki Onodera.

**Validation:** Hideki Hirakawa, Yasuyuki Onodera.

**Visualization:** Yasuyuki Onodera.

**Writing – original draft:** Keisuke Tanaka, Hideki Hirakawa, Yasuyuki Onodera.

**Writing – review & editing:** Yasuyuki Onodera.

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
