## [Decision Letter · Decision Letter 0]

20 Oct 2023

PONE-D-23-25608QTL analysis of femaleness in monoecious spinach and fine mapping of a major QTL using an updated version of chromosome-scale pseudomoleculesPLOS ONE

Dear Dr. Onodera,

Thank you for submitting your manuscript to PLOS ONE. After careful consideration, we feel that it has merit but does not fully meet PLOS ONE’s publication criteria as it currently stands. Therefore, we invite you to submit a revised version of the manuscript that addresses the points raised during the review process.

We look forward to receiving your revised manuscript.

Kind regards,

Mojtaba Kordrostami, Ph.D.

Academic Editor

PLOS ONE

Journal Requirements:

Additional Editor Comments:

Dear Authors,

Thank you for submitting your manuscript to PLOS ONE. We have received feedback from two expert reviewers, and after careful consideration, we have decided that major revisions are required before your manuscript can be considered for publication.

Please find below the comments from the reviewers:

Reviewer 1:

The manuscript is well done regarding both practical and analytical approaches and is presented in a good scientific format.

It is recommended to check the manuscript by a native English editor for grammatical correctness.

Reviewer 2:

Abstract:

Clarify the significance of the SOL_r2.0_pseudomolecule in comparison to previous versions or other existing pseudomolecules for spinach.

Provide more context on the importance of understanding the genetic basis of monoecious expression in spinach.

Elaborate on the methods used to determine the high gene completeness (BUSCO complete 97.0%) of the SOL_r2.0_pseudomolecule.

Specify the criteria used to determine the LOD score and explain why qFem3.1 had the highest LOD score in comparison to the other QTLs.

Provide more information on the M locus and its implications in spinach genetics.

Elaborate on the significance of the RADIALIS-like-2a (SoRL2a) gene and its association with reproductive development in spinach.

Provide more details on the genes associated with floral organogenesis and auxin transport/response identified from the chromosomal region for qFem2.1 and qFem6.1.

Discuss the potential applications of your findings in breeding practices.

Consider rephrasing certain sentences for clarity and conciseness.

Introduction:

Provide a brief overview of the significance of spinach in global agriculture.

Elaborate on the advantages and disadvantages of dioecious versus monoecious plants in spinach cultivation.

Clarify the significance of the X and Y chromosomes in determining sexual dimorphism in spinach.

Provide more context on the importance of the M locus in relation to other loci influencing the monoecious phenotype in spinach.

Discuss the implications of previous genetic analysis findings.

Explain the significance of high-quality genome assembly or chromosome-scale pseudomolecules in plant genetics and breeding.

Elaborate on the improvements made in the SOL_r2.0_pseudomolecule compared to the SOL_r1.0_pseudomolecule.

Discuss the potential applications of the findings in breeding practices.

Consider rephrasing certain sentences for clarity and conciseness.

Material and Methods:

Provide a rationale for the choice of specific spinach lines used in the study.

Clarify the significance of the near-isogenic line NIL-M.

Elaborate on the importance of the Hi-C technique in improving the accuracy of the pseudomolecules.

Provide more details on the criteria used for QTL analysis.

Discuss the implications of the identified nucleotide variants.

Elaborate on the significance of the in situ hybridization technique.

Provide more context on the importance of GO enrichment analysis.

Consider rephrasing certain sentences for clarity and conciseness.

Results:

Provide a brief background on the significance of constructing linkage maps and pseudomolecules for plant genetics and breeding.

Clarify the importance of updating the set of pseudomolecules for spinach line 03-009.

Explain the relevance of the N50 value and BUSCO completeness in assessing the quality of the genome assembly.

Provide more context on the significance of anchoring scaffolds to linkage groups and the implications of trimming based on the M/F ratio score.

Consider rephrasing certain sentences for clarity and conciseness.

Discussion:

Provide more context on why the SOL_r2.0_pseudomolecule is a significant improvement over the previous version, SOL_r1.1.

Clarify the advantages and limitations of using long-read sequencing technologies combined with Hi-C proximity data.

Elaborate on the potential applications of the SOL_r2.0_pseudomolecule as a reference genome.

Provide more details on the methodology used to identify the three QTLs regulating femaleness in monoecious spinach plants.

Discuss the potential implications of the identified QTLs in practical breeding programs.

Discuss the potential functional implications of the SoRL2a gene in the context of floral asymmetry.

Provide more evidence or references supporting the role of non-coding RNAs in monoecious expression in spinach.

Delve deeper into the role of auxins in sex differentiation.

Discuss the potential significance of the floral-organogenesis-related and auxin-related genes in the qfem2.1 and qfem6.1 regions.

Consider rephrasing certain sentences for clarity and conciseness.

Please address these comments and revise your manuscript accordingly. Once you have made the necessary changes, please resubmit your manuscript along with a detailed response to each of the reviewer's comments, explaining how you have addressed their concerns.

Thank you for considering PLOS ONE as a platform for your research. We look forward to receiving your revised manuscript.

Best regards,

Mojtaba Kordrostami

Editor

PLOS ONE

Reviewers' comments:

Reviewer's Responses to Questions

**Comments to the Author**

1. Is the manuscript technically sound, and do the data support the conclusions?

Reviewer #1: Yes

Reviewer #2: No

2. Has the statistical analysis been performed appropriately and rigorously? 

Reviewer #1: Yes

Reviewer #2: No

3. Have the authors made all data underlying the findings in their manuscript fully available?

Reviewer #1: Yes

Reviewer #2: Yes

4. Is the manuscript presented in an intelligible fashion and written in standard English?

Reviewer #1: Yes

Reviewer #2: No

5. Review Comments to the Author

Reviewer #1: from the reviewer point view, this research work were done well regarding both practical and analytical approach and also presented in good scientific format. Just recommended to check the manuscript by native English editor based on grammatical rules.

Reviewer #2: Abstract:

1. The authors should clarify the significance of the SOL_r2.0_pseudomolecule in the context of previous versions or other existing pseudomolecules for spinach. How does this version compare or improve upon previous ones?

2. The authors should provide more context on the importance of understanding the genetic basis of monoecious expression in spinach. Why is this trait significant for breeding or agricultural practices?

3. The authors should elaborate on the methods used to determine the high gene completeness (BUSCO complete 97.0%) of the SOL_r2.0_pseudomolecule.

4. The authors should specify the criteria used to determine the LOD score and explain why qFem3.1 had the highest LOD score in comparison to the other QTLs.

5. The authors should provide more information on the M locus. How was it previously identified as the major locus responsible for monoecious expression? What are the implications of this locus in the broader context of spinach genetics?

6. The authors should elaborate on the significance of the RADIALIS-like-2a (SoRL2a) gene. How does its expression pattern associated with reproductive development contribute to the monoecious expression in spinach?

7. The authors should provide more details on the genes associated with floral organogenesis and auxin transport/response identified from the chromosomal region for qFem2.1 and qFem6.1. How do these genes influence the ratio of female to male flowers in monoecious plants?

8. The authors should discuss the potential applications of their findings in breeding practices. How can breeders utilize this information to develop new spinach varieties?

9. The authors should consider rephrasing the sentence "Spinach is generally considered a dioecious vegetable, but monoecious plants are found in certain lines, cultivars, and genotypes, with various proportions of female and male flowers." to "Although spinach is predominantly dioecious, certain lines, cultivars, and genotypes exhibit monoecious plants with varying proportions of female and male flowers."

10. The authors should replace "on which" with "wherein" in the sentence "We here present SOL_r2.0_pseudomolecule, a set of six pseudomolecules (879.2 Mb in total length) covering most parts of the spinach genome with high gene completeness (BUSCO complete 97.0%), on which three QTLs responsible for the expression of monoecious characters were mapped;"

11. The authors should consider rephrasing "Our fine-mapping efforts narrowed down the candidate region for the M locus to a 19.5 kb interval harboring three protein-coding genes and a long non-coding RNA gene." to "Through fine-mapping, we narrowed the candidate region for the M locus to a 19.5 kb interval that contains three protein-coding genes and one long non-coding RNA gene."

12. The authors should replace "highly male and female monoecious plants" with "monoecious plants with a high proportion of male or female flowers" in the sentence "Our findings will help breeders to efficiently select highly male and female monoecious plants suited to the paternal and maternal parents of F1 hybrids."

Introduction:

1. The authors should provide a brief overview of the significance of spinach in global agriculture and its nutritional or economic importance to set the context for readers unfamiliar with the crop.

2. The authors should elaborate on the advantages and disadvantages of dioecious versus monoecious plants in spinach cultivation. This will help readers understand the importance of studying the genetic basis of these traits.

3. The authors should clarify the significance of the X and Y chromosomes in determining sexual dimorphism in spinach. How do these findings compare to other plant species with similar sexual determination systems?

4. The authors should provide more context on the importance of the M locus in relation to other loci that might influence the monoecious phenotype in spinach.

5. The authors should discuss the implications of the findings from the previous genetic analysis, especially the relationship between the Y, M, and X genes and their roles in determining plant sex.

6. The authors should explain the significance of high-quality genome assembly or chromosome-scale pseudomolecules in the broader context of plant genetics and breeding.

7. The authors should elaborate on the improvements made in the SOL_r2.0_pseudomolecule compared to the SOL_r1.0_pseudomolecule. What were the challenges with the previous version, and how were they addressed in the updated version?

8. The authors should discuss the potential applications of their findings in breeding practices and how they can be used to develop new spinach varieties with desired traits.

9. The authors should consider rephrasing the sentence "Spinach is considered to have originated in Persia and has since been widely introduced in East Asia, Europe, and the Americas;" to "Originating from Persia, spinach has since been introduced widely across East Asia, Europe, and the Americas;"

10. The authors should replace "In the earliest days of spinach hybrid breeding programs," with "In the early stages of spinach hybrid breeding," for clarity and conciseness.

11. The authors should consider rephrasing "For this reason, the most critical issues in spinach hybrid breeding programs are elucidating the mechanisms underlying the sex determination in dioecious lines, and controlling the degree of femaleness (the percentage of female flowers per plant) in monoecious spinach." to "Therefore, understanding the mechanisms of sex determination in dioecious lines and controlling the degree of femaleness in monoecious spinach are critical issues in hybrid breeding programs."

12. The authors should replace "That is, the chromosome carrying the dominant male determinant (Y) is the Y chromosome, and the other one is the X chromosome" with "Specifically, the chromosome with the dominant male determinant is the Y chromosome, while its counterpart is the X chromosome."

13. The authors should consider rephrasing "High-quality genome assembly or chromosome-scale pseudomolecules enable us to efficiently identify genes and loci of interest and provide a basis for genome-based molecular breeding strategies—e.g., marker-assisted selection and genomic selection [15]." to "High-quality genome assemblies, such as chromosome-scale pseudomolecules, facilitate the identification of genes and loci of interest, laying the foundation for genome-based molecular breeding strategies like marker-assisted and genomic selection."

14. The authors should replace "Here, we improved the length and accuracy of the pseudomolecules by using Hi-C," with "In this study, we enhanced the length and accuracy of the pseudomolecules using Hi-C," for clarity.

Material and Methods:

1. The authors should provide a brief rationale for the choice of the specific spinach lines (03-009 and 03-336) used in this study. What makes these lines particularly suitable for the research objectives?

2. The authors should clarify the significance of the near-isogenic line NIL-M. How does this line differ from the parent lines, and what advantages does it offer for the study?

3. The authors should elaborate on the importance of the Hi-C technique in improving the accuracy of the pseudomolecules. How does Hi-C contribute to the understanding of genome architecture and the identification of QTLs?

4. The authors should provide more details on the criteria used for QTL analysis, specifically the thresholds or parameters used to determine significant QTLs.

5. The authors should discuss the implications of the identified nucleotide variants. How do these variants influence the traits of interest, and what is their significance in the broader context of spinach genetics?

6. The authors should elaborate on the significance of the in situ hybridization technique. How does this method contribute to the understanding of gene expression patterns in the studied tissues?

7. The authors should provide more context on the importance of GO enrichment analysis. How does this analysis help in understanding the functional roles of the differentially expressed genes?

8. The authors should consider rephrasing "The spinach F2 population (03-009 x 03-336 F2; [16]) was produced by a cross between a female plant (mm plant) of line 03-009 and a plant (MM plant) from the highly male monoecious line 03-336" to "The spinach F2 population (03-009 x 03-336 F2; [16]) resulted from crossing a female plant (mm plant) of line 03-009 with a plant (MM plant) from the highly male monoecious line 03-336."

9. The authors should replace "Plants used for preliminary RNA-seq, RT-qPCR, and QTL analyses were grown in a growth chamber" with "For preliminary RNA-seq, RT-qPCR, and QTL analyses, plants were cultivated in a growth chamber."

10. The authors should consider rephrasing "Plants used for verifying QTL effects and transcriptome analysis were grown at a constant temperature of 20°C" to "For QTL verification and transcriptome analysis, plants were maintained at a consistent 20°C."

11. The authors should replace "In this study, plants bearing more than 20 flower clusters, each consisting of 5–10 flowers, were evaluated for femaleness." with "For this study, we evaluated the femaleness of plants that bore over 20 flower clusters, with each cluster comprising 5–10 flowers."

12. The authors should consider rephrasing "The RNA seq libraries used for preliminary expression analysis were prepared from the RNA samples according to a method described in Takahata et al. [13]" to "Using the method described in Takahata et al. [13], we prepared RNA seq libraries from the RNA samples for preliminary expression analysis."

13. The authors should replace "RNA seq libraries for the main expression analysis were prepared using a NEBNext Ultra II Directional RNA Library Prep Kit" with "For the primary expression analysis, we prepared RNA seq libraries using the NEBNext Ultra II Directional RNA Library Prep Kit."

Results

Construction of a linkage map and chromosome-scale pseudomolecules for spinach line 03-009.

1. The authors should provide a brief background on the significance of constructing linkage maps and pseudomolecules for plant genetics and breeding.

2. The authors should clarify the importance of updating the set of pseudomolecules (SOL_r1.0_pseudomolecule) for spinach line 03-009.

3. The authors should explain the relevance of the N50 value and BUSCO completeness in assessing the quality of the genome assembly.

4. The authors should provide more context on the significance of anchoring scaffolds to linkage groups and the implications of trimming based on the M/F ratio score.

5. The authors should consider rephrasing "polished with long and short reads" to "refined using both long and short reads" for clarity.

6. The authors should ensure that all terms and abbreviations are consistently capitalized, such as "Linkage analysis" and "linkage groups."

Fine mapping of the M locus and expression analysis of its candidate genes.

1. The authors should provide a brief introduction to the significance of the M locus in spinach and its role in determining plant sex.

2. The authors should explain the importance of fine mapping in identifying candidate genes and their potential functions.

3. The authors should discuss the implications of identifying SoRL2a as a promising candidate gene for male organogenesis and monoecious expression.

4. The authors should consider rephrasing "the locus for monoecism (M) was located between" to "the locus responsible for monoecism (M) was positioned between."

5. The authors should ensure that all figures and tables are consistently referenced, such as "Fig 2" and "Figure 2."

Mapping of QTLs controlling femaleness in monoecious spinach.

1. The authors should provide a brief background on the significance of QTL mapping in plant genetics and its role in identifying genes associated with specific traits.

2. The authors should discuss the implications of identifying qFem2.1, qFem3.1, and qFem6.1 as QTLs controlling femaleness in spinach.

3. The authors should consider rephrasing "female plants (mm) from line 03-009 and monoecious plants (MM) from line 03-336" to "female plants of genotype mm from line 03-009 and monoecious plants of genotype MM from line 03-336" for clarity.

4. The authors should ensure that all terms and abbreviations are consistently capitalized, such as "QTL analysis" and "QTLs."

Discussion:

1. Genome Assembly and Comparison:

o The authors should provide more context on why the SOL_r2.0_pseudomolecule is a significant improvement over the previous version, SOL_r1.1. Specifically, they should quantify the reduction in scaffolding errors and other improvements.

o The authors should clarify the advantages and limitations of using long-read sequencing technologies combined with Hi-C proximity data over other methods like mate-pair reads.

2. Reference Genome:

o The authors should elaborate on the potential applications of the SOL_r2.0_pseudomolecule as a reference genome, especially in comparison to other available spinach genomes.

3. QTLs and Genetic Mapping:

o The authors should provide more details on the methodology used to identify the three QTLs regulating femaleness in monoecious spinach plants.

o The authors should discuss the potential implications of the identified QTLs, especially qFem3.1, in practical breeding programs.

4. Gene Expression and Function:

o The authors should discuss the potential functional implications of the SoRL2a gene in the context of floral asymmetry and its comparison with other homologs like DkRAD.

o The authors should provide more evidence or references supporting the role of non-coding RNAs in the monoecious expression in spinach.

5. Auxins and Sex Differentiation:

o The authors should delve deeper into the role of auxins in sex differentiation, providing more evidence from the literature and discussing the potential mechanisms at play.

o The authors should discuss the potential significance of the floral-organogenesis-related and auxin-related genes in the qfem2.1 and qfem6.1 regions.

1. In the sentence "Considering that the spinach genome is relatively large (~1000 Mb) and rich with repetitive elements...", the authors should consider rephrasing to "Given that the spinach genome is relatively large (~1000 Mb) and abundant in repetitive elements..."

2. In the sentence "Monoe-Viroflay and Viroflay are likely derived...", the authors should consider changing "are likely derived" to "likely derive".

3. In the sentence "By contrast, breeding line 03-009 originated from a Japanese cultivar.", the authors should consider rephrasing to "In contrast, the breeding line 03-009 originates from a Japanese cultivar."

4. In the sentence "Our research successfully identified three QTLs regulating femaleness...", the authors should consider changing "Our research" to "This study".

5. In the sentence "Among three protein-coding genes in the M candidate region...", the authors should consider rephrasing to "Of the three protein-coding genes in the M candidate region..."

6. In the sentence "Sex differentiation of the spinach flowers is not involved...", the authors should consider rephrasing to "Sex differentiation in spinach flowers does not involve..."

7. The authors should ensure consistent use of tense throughout the discussion. For instance, they switch between past and present tense, which can be confusing for readers.

8. The authors should consider breaking up longer sentences into shorter, more concise statements for clarity.

9. The authors should ensure that all references are correctly formatted and consistent throughout the discussion.

6. PLOS authors have the option to publish the peer review history of their article (what does this mean?). If published, this will include your full peer review and any attached files.

Reviewer #1: No

Reviewer #2: No

---

## [Author Response · Author response to Decision Letter 0]

30 Nov 2023

Responses to Reviewer 1:

General comment: from the reviewer point view, this research work were done well regarding both practical and analytical approach and also presented in good scientific format. Just recommended to check the manuscript by native English editor based on grammatical rules.

Response: We have had the entire manuscript revised by an expert whose native language is English. A certificate of proofreading is attached.

Responses to Reviewer 2:

Abstract:

Comment 1. The authors should clarify the significance of the SOL_r2.0_pseudomolecule in the context of previous versions or other existing pseudomolecules for spinach. How does this version compare or improve upon previous ones?

Response: Lines 37–41. We added a passage to provide this information. Thank you for the suggestion. 

Comment 2: The authors should provide more context on the importance of understanding the genetic basis of monoecious expression in spinach. Why is this trait significant for breeding or agricultural practices?

Response: Lines 33–37. We revised the manuscript to provide more context on this point. 

Comment 3: The authors should elaborate on the methods used to determine the high gene completeness (BUSCO complete 97.0%) of the SOL_r2.0_pseudomolecule.

Response. Lines 37–41. We revised the manuscript in accordance with your suggestion. However, to maintain the length of the Abstract at 300 words, we provided these details in the Materials and Methods, and Results sections instead (lines 226–229, lines 319–324).

Comment 4: The authors should specify the criteria used to determine the LOD score and explain why qFem3.1 had the highest LOD score in comparison to the other QTLs.

Response: Lines 41–45. We added this information as suggested. Here again though, we decided to provide this information in the Materials and Methods in order to hold the Abstract to 300 words (lines 213–217).

Comment 5: The authors should provide more information on the M locus. How was it previously identified as the major locus responsible for monoecious expression? What are the implications of this locus in the broader context of spinach genetics?

Response: Lines 42–45. We added this information as recommended. 

Comment 6: The authors should elaborate on the significance of the RADIALIS-like-2a (SoRL2a) gene. How does its expression pattern associated with reproductive development contribute to the monoecious expression in spinach?

Response: Lines 50–54. We added a passage to address these important questions. 

Comment 7: The authors should provide more details on the genes associated with floral organogenesis and auxin transport/response identified from the chromosomal region for qFem2.1 and qFem6.1. How do these genes influence the ratio of female to male flowers in monoecious plants?

Response: Lines 54–57. We revised the manuscript in accordance with your suggestion. 

Comment 8: The authors should discuss the potential applications of their findings in breeding practices. How can breeders utilize this information to develop new spinach varieties?

Response. Lines 46–48. We added a passage discussing the potential applications. Thank you for the suggestion. 

Comment 9: The authors should consider rephrasing the sentence “Spinach is generally considered a dioecious vegetable, but monoecious plants are found in certain lines, cultivars, and genotypes, with various proportions of female and male flowers.” to “Although spinach is predominantly dioecious, certain lines, cultivars, and genotypes exhibit monoecious plants with varying proportions of female and male flowers.”

Response: Lines 33-34. Thank you for the revision. We adopted the revised sentence. 

Comment 10: The authors should replace “on which” with “wherein” in the sentence “We here present SOL_r2.0_pseudomolecule, a set of six pseudomolecules (879.2 Mb in total length) covering most parts of the spinach genome with high gene completeness (BUSCO complete 97.0%), on which three QTLs responsible for the expression of monoecious characters were mapped;”

Response: Line 41. We revised it so that it is written in two sentences.

Comment 11: The authors should consider rephrasing “Our fine-mapping efforts narrowed down the candidate region for the M locus to a 19.5 kb interval harboring three protein-coding genes and a long non-coding RNA gene.” to “Through fine-mapping, we narrowed the candidate region for the M locus to a 19.5 kb interval that contains three protein-coding genes and one long non-coding RNA gene.”

Response: Lines 48-50. We changed the sentence as recommended.

Comment 12: The authors should replace “highly male and female monoecious plants” with “monoecious plants with a high proportion of male or female flowers” in the sentence “Our findings will help breeders to efficiently select highly male and female monoecious plants suited to the paternal and maternal parents of F1 hybrids.”

Response: Lines 46–48. Thank you for the recommendation. We changed the sentence as recommended, but with the addition of some relevant information as follows: “Our findings will enable breeders to efficiently produce highly female- and male-monoecious parental lines for F1-hybrids by pyramiding the three QTLs.”

Introduction:

Comment 1: The authors should provide a brief overview of the significance of spinach in global agriculture and its nutritional or economic importance to set the context for readers unfamiliar with the crop.

Response: Lines 61–64. We added an overview as suggested.

Comment 2: The authors should elaborate on the advantages and disadvantages of dioecious versus monoecious plants in spinach cultivation. This will help readers understand the importance of studying the genetic basis of these traits.

Response: Lines 74–80. We revised the manuscript in accordance with your suggestion.

Comment 3: The authors should clarify the significance of the X and Y chromosomes in determining sexual dimorphism in spinach. How do these findings compare to other plant species with similar sexual determination systems?

Response: Lines 81–100. We added sentences to clarify this point as recommended.

Comment 4: The authors should provide more context on the importance of the M locus in relation to other loci that might influence the monoecious phenotype in spinach.

Response: Lines 117–123. Thank you for the suggestion. We added additional context. 

Comment 5: The authors should discuss the implications of the findings from the previous genetic analysis, especially the relationship between the Y, M, and X genes and their roles in determining plant sex.

Response: Lines 100–116. We revised the manuscript in accordance with your suggestion.

Comment 6: The authors should explain the significance of high-quality genome assembly or chromosome-scale pseudomolecules in the broader context of plant genetics and breeding.

Response: Lines 124–131. We added the recommended explanation.

Comment 7: The authors should elaborate on the improvements made in the SOL_r2.0_pseudomolecule compared to the SOL_r1.0_pseudomolecule. What were the challenges with the previous version, and how were they addressed in the updated version?

Response: Lines 135–144. We revised the manuscript in accordance with your suggestion. However, since this content was also provided in the Discussion section (Lines 702–719), we added a shorter text here.

Comment 8: The authors should discuss the potential applications of their findings in breeding practices and how they can be used to develop new spinach varieties with desired traits.

Response: Lines 737–742. We added a sentence to the Discussion section to address potential applications.

Comment 9: The authors should consider rephrasing the sentence “Spinach is considered to have originated in Persia and has since been widely introduced in East Asia, Europe, and the Americas;” to “Originating from Persia, spinach has since been introduced widely across East Asia, Europe, and the Americas;”

Response: Lines 60–61. Thank you for your close reading. We adopted the new version of the sentence.

Comment 10: The authors should replace “In the earliest days of spinach hybrid breeding programs,” with “In the early stages of spinach hybrid breeding,” for clarity and conciseness.

Response: Lines 68–69. Again, we appreciate your close attention to the syntax. We revised to the more concise phrase as suggested.

Comment 11: The authors should consider rephrasing “For this reason, the most critical issues in spinach hybrid breeding programs are elucidating the mechanisms underlying the sex determination in dioecious lines, and controlling the degree of femaleness (the percentage of female flowers per plant) in monoecious spinach.” to “Therefore, understanding the mechanisms of sex determination in dioecious lines and controlling the degree of femaleness in monoecious spinach are critical issues in hybrid breeding programs.”

Response: Lines 78–80. We revised the sentence as recommended.

Comment 12: The authors should replace “That is, the chromosome carrying the dominant male determinant (Y) is the Y chromosome, and the other one is the X chromosome” with “Specifically, the chromosome with the dominant male determinant is the Y chromosome, while its counterpart is the X chromosome.”

Response: Lines 98–100. Thank you for the rewrite. We have adopted the new sentence.

Comment 13: The authors should consider rephrasing “High-quality genome assembly or chromosome-scale pseudomolecules enable us to efficiently identify genes and loci of interest and provide a basis for genome-based molecular breeding strategies—e.g., marker-assisted selection and genomic selection [15].” to “High-quality genome assemblies, such as chromosome-scale pseudomolecules, facilitate the identification of genes and loci of interest, laying the foundation for genome-based molecular breeding strategies like marker-assisted and genomic selection.”

Response: Lines 124–131. The points you pointed out have been revised as follows to include additional information:　

Having high-quality genome assemblies, such as chromosome-scale pseudomolecules, is crucial for identifying genes and loci of interest. Such pseudomolecules are particularly important as a reference because GWAS and QTL analyses require highly accurate variant data that covers nearly the whole genome. The pseudomolecules can be constructed by aligning highly accurate scaffolds in the appropriate direction and have high genome coverage rates. 

Comment 14: The authors should replace “Here, we improved the length and accuracy of the pseudomolecules by using Hi-C,” with “In this study, we enhanced the length and accuracy of the pseudomolecules using Hi-C,” for clarity.

Response: Lines 140–143. The following changes have been made to provide more detailed information:

Using pseudomolecules with a genome coverage rate of about 70% as a reference for GWAS and QTL analyses, there is a risk that the genes of interest may not be detected and identified. Therefore, in this study, we enhanced the length and accuracy of the pseudomolecules using Hi-C, …

Materials and Methods:

Comment 1: The authors should provide a brief rationale for the choice of the specific spinach lines (03-009 and 03-336) used in this study. What makes these lines particularly suitable for the research objectives?

Response: Lines 150–158. We added a brief explanation of our reasons for choosing these particular spinach lines.

Comment 2: The authors should clarify the significance of the near-isogenic line NIL-M. How does this line differ from the parent lines, and what advantages does it offer for the study?

Response: Lines 166–168. We added a sentence to clarify the significance of NIL-M. 

Comment 3: The authors should elaborate on the importance of the Hi-C technique in improving the accuracy of the pseudomolecules. How does Hi-C contribute to the understanding of genome architecture and the identification of QTLs?

Response: Lines 187–192. We added a passage discussing the importance of the Hi-C technique.

Comment 4: The authors should provide more details on the criteria used for QTL analysis, specifically the thresholds or parameters used to determine significant QTLs.

Response: Lines 213–217. We added these details as suggested.

Comment 5: The authors should discuss the implications of the identified nucleotide variants. How do these variants influence the traits of interest, and what is their significance in the broader context of spinach genetics?

Response: Lines 239–246: Thank you for the suggestion. We added this information.

Comment 6: The authors should elaborate on the significance of the in situ hybridization technique. How does this method contribute to the understanding of gene expression patterns in the studied tissues?

Response: Lines 265–266. We added a passage to address the significance of this technique.

Comment 7: The authors should provide more context on the importance of GO enrichment analysis. How does this analysis help in understanding the functional roles of the differentially expressed genes?

Response: Lines 302–304. We added the recommended context.

Comment 8: The authors should consider rephrasing “The spinach F2 population (03-009 x 03-336 F2; [16]) was produced by a cross between a female plant (mm plant) of line 03-009 and a plant (MM plant) from the highly male monoecious line 03-336” to “The spinach F2 population (03-009 x 03-336 F2; [16]) resulted from crossing a female plant (mm plant) of line 03-009 with a plant (MM plant) from the highly male monoecious line 03-336.”

Response: Lines 156–158. We changed the sentence as recommended.

Comment 9: The authors should replace “Plants used for preliminary RNA-seq, RT-qPCR, and QTL analyses were grown in a growth chamber” with “For preliminary RNA-seq, RT-qPCR, and QTL analyses, plants were cultivated in a growth chamber.”

Response: Lines 169–170. We changed the sentence as recommended.

Comment 10: The authors should consider rephrasing “Plants used for verifying QTL effects and transcriptome analysis were grown at a constant temperature of 20°C” to “For QTL verification and transcriptome analysis, plants were maintained at a consistent 20°C.”

Response: Lines 172–173. We rephrased this material as recommended.

Comment 11: The authors should replace “In this study, plants bearing more than 20 flower clusters, each consisting of 5–10 flowers, were evaluated for femaleness.” with “For this study, we evaluated the femaleness of plants that bore over 20 flower clusters, with each cluster comprising 5–10 flowers.”

Response: Lines 175–176. We adopted the revised version of the sentence.

Comment 12: The authors should consider rephrasing “The RNA seq libraries used for preliminary expression analysis were prepared from the RNA samples according to a method described in Takahata et al. [13]” to “Using the method described in Takahata et al. [13], we prepared RNA seq libraries from the RNA samples for preliminary expression analysis.”

Response: Lines 281–282. We changed the sentence as recommended.

Comment 13: The authors should replace “RNA seq libraries for the main expression analysis were prepared using a NEBNext Ultra II Directional RNA Library Prep Kit” with “For the primary expression analysis, we prepared RNA seq libraries using the NEBNext Ultra II Directional RNA Library Prep Kit.”

Response: Lines 289–290. We changed the sentence as recommended.

Results: Construction of a linkage map and chromosome-scale pseudomolecules for spinach line 03-009:

Comment 1: The authors should provide a brief background on the significance of constructing linkage maps and pseudomolecules for plant genetics and breeding.

Response: Lines 315–319. We added the information as recommended. Since this content is also described in the Introduction section, we used a shortened version here.

Comment 2: The authors should clarify the importance of updating the set of pseudomolecules (SOL_r1.0_pseudomolecule) for spinach line 03-009.

Response: Lines 140–144. We added this information in the Introduction section instead of the Results section.

Comment 3: The authors should clarify the importance of updating the set of pseudomolecules (SOL_r1.0_pseudomolecule) for spinach line 03-009.

Response: Lines 315–319. We added sentences to clarify this point. We also provided this information in the Introduction section (Lines 124–142).

Comment 4. The authors should explain the relevance of the N50 value and BUSCO completeness in assessing the quality of the genome assembly.

Response: Lines 322–324. We added this information as recommended.

Comment 5: The authors should provide more context on the significance of anchoring scaffolds to linkage groups and the implications of trimming based on the M/F ratio score.

Response: Lines 329–340. We added additional context.

Comment 6: The authors should consider rephrasing “polished with long and short reads” to “refined using both long and short reads” for clarity.

Response: Lines 318–319. Thank you for the suggestion. We adopted the revised phrasing. 

Comment 7: The authors should ensure that all terms and abbreviations are consistently capitalized, such as “Linkage analysis” and “linkage groups.” 

Response: We checked all the capitalizations for consistency. We used lowercase for “linkage analysis” and “linkage groups”, as they are not abbreviations or proper nouns, except when appearing at the beginning of a sentence. 

Fine mapping of the M locus and expression analysis of its candidate genes:

Comment 1: The authors should provide a brief introduction to the significance of the M locus in spinach and its role in determining plant sex.

Response: Lines 372–373. We added a brief introduction as recommended.

Comment 2: The authors should explain the importance of fine mapping in identifying candidate genes and their potential functions.

Response: Lines 378–389. We added the suggested explanation.

Comment 3: The authors should discuss the implications of identifying SoRL2a as a promising candidate gene for male organogenesis and monoecious expression.

Response: Lines 480–488. We added this information as recommended.

Comment 4: The authors should consider rephrasing “the locus for monoecism (M) was located between” to “the locus responsible for monoecism (M) was positioned between.”

Response: Lines 372–373. We used the recommended phrasing. 

Comment 5: The authors should ensure that all figures and tables are consistently referenced, such as “Fig 2” and “Figure 2.”

Response: We checked the consistency of all references to figures and tables.

Mapping of QTLs controlling femaleness in monoecious spinach:

Comment 1: The authors should provide a brief background on the significance of QTL mapping in plant genetics and its role in identifying genes associated with specific traits.

Response: Lines 569–570. To clarify the purpose of QTL analysis, we rewrote the relevant sentence as follows: “In order to identify the additional loci for monoecism, other than M, QTL analysis for femaleness was performed using the 03-336 x 03-009 F2 population.”

Comment 2: The authors should discuss the implications of identifying qFem2.1, qFem3.1, and qFem6.1 as QTLs controlling femaleness in spinach.

Response: As mentioned above, to make it clear that the additional loci (qFem2.1 and qFem6.1) were successfully identified, we rewrote the relevant sentence as follows: “In order to identify the additional loci for monoecism, other than M, QTL analysis for femaleness was performed using the 03-336 x 03-009 F2 population.” The roles of qFem2.1, qFem3.1 (M), and qFem6.1 in controlling femaleness are described in the subsection qFem2.1 and qFem6.1 act as modifiers of the M gene function. Additionally, we added information about the role of qFem2.1, qFem3.1, and qFem6.1 in controlling femaleness and corrected the corresponding sentences in the Discussion section (Lines 755–758). 

Comment 3: The authors should consider rephrasing “female plants (mm) from line 03-009 and monoecious plants (MM) from line 03-336” to “female plants of genotype mm from line 03-009 and monoecious plants of genotype MM from line 03-336” for clarity.

Response: Lines 534–535. We used the suggested phrasing.

Comment 4: The authors should ensure that all terms and abbreviations are consistently capitalized, such as “QTL analysis” and “QTLs.”

Response: We revised the terminology throughout the paper for consistency. However, since “QTLs” is widely used in genetics papers as the plural of “quantitative trait locus”, we decided to leave this acronym as is.

Discussion:

Genome Assembly and Comparison:

Comment 1: The authors should provide more context on why the SOL_r2.0_pseudomolecule is a significant improvement over the previous version, SOL_r1.1. Specifically, they should quantify the reduction in scaffolding errors and other improvements.

Response: Lines 702–705, 715–719. We made the suggested changes.

Comment 2: The authors should clarify the advantages and limitations of using long-read sequencing technologies combined with Hi-C proximity data over other methods like mate-pair reads.

Response: Lines 705–713. We added a passage to clarify the advantages and limitations.

Reference Genome:

Comment 1: The authors should elaborate on the potential applications of the SOL_r2.0_pseudomolecule as a reference genome, especially in comparison to other available spinach genomes.

Response: Lines 737–742. We made the suggested revisions.

QTLs and Genetic Mapping:

Comment 1: The authors should provide more details on the methodology used to identify the three QTLs regulating femaleness in monoecious spinach plants.

Response: Lines 744–746. We added this information as suggested. Since this content was also provided in the Results section (Lines 528–575), we used a shortened version here.

Comment 2: The authors should discuss the potential implications of the identified QTLs, especially qFem3.1, in practical breeding programs.

Response: Lines 761–765. We added information about the implications of the QTLs in practical breeding programs

.

Gene Expression and Function:

Comment 1: The authors should discuss the potential functional implications of the SoRL2a gene in the context of floral asymmetry and its comparison with other homologs like DkRAD.

Response: Lines765–801. As described in the manuscript (lines 766–781), the roles of the RADIALIS family genes appear to be diverse. Based on this, we revised the manuscript with the suggested discussion lines (lines 793–802). 

Comment 2: The authors should provide more evidence or references supporting the role of non-coding RNAs in the monoecious expression in spinach.

Response: Lines 803–817. We added the recommended material. 

Auxins and Sex Differentiation:

Comment 1: The authors should delve deeper into the role of auxins in sex differentiation, providing more evidence from the literature and discussing the potential mechanisms at play.

Response: Lines 865–878. We provided additional information on the role of auxins in sex differentiation. 

Comment 2: The authors should discuss the potential significance of the floral-organogenesis-related and auxin-related genes in the qfem2.1 and qfem6.1 regions.

Response: Lines 886–900. We added sentences to address this point. 

Comment 3: In the sentence “Considering that the spinach genome is relatively large (~1000 Mb) and rich with repetitive elements...”, the authors should consider rephrasing to “Given that the spinach genome is relatively large (~1000 Mb) and abundant in repetitive elements...”

Response: Lines 705–707. Due to major additions, we have revised it as follows: 

This may have been due to the size of the spinach genome, which is relatively large (~1000 Mb) and contains many repetitive elements. Additionally, the mate-pair libraries used to construct SOL_r1.1 were limited to 3, 6, 10, and 15 kb inserts,…

Comment 4: In the sentence “Monoe-Viroflay and Viroflay are likely derived...”, the authors should consider changing “are likely derived” to “likely derive”.

Response: Line 726. We made the suggested change.

Comment 5: In the sentence “By contrast, breeding line 03-009 originated from a Japanese cultivar.”, the authors should consider rephrasing to “In contrast, the breeding line 03-009 originates from a Japanese cultivar.”

Response: Lines 729–730. We changed the sentence as suggested.

Comment 6: In the sentence “Our research successfully identified three QTLs regulating femaleness...”, the authors should consider changing “Our research” to “This study”.

Response: Line 743. We changed the phrasing as suggested.

Comment 7: In the sentence “Among three protein-coding genes in the M candidate region...”, the authors should consider rephrasing to “Of the three protein-coding genes in the M candidate region...”

Response: Line 766. We changed the phrasing as suggested.

Comment 8: In the sentence “Sex differentiation of the spinach flowers is not involved...”, the authors should consider rephrasing to “Sex differentiation in spinach flowers does not involve...”

Response: Line 782. We changed the phrasing as suggested.

Comment 9: The authors should ensure consistent use of tense throughout the discussion. For instance, they switch between past and present tense, which can be confusing for readers.

Response: We have had the entire manuscript revised by an expert whose native language is English. A certificate of proofreading is attached.

Comment 10: The authors should consider breaking up longer sentences into shorter, more concise statements for clarity.

Response: We reviewed the entire text with an eye toward shortening sentences where possible. 

Comment 11: The authors should ensure that all references are correctly formatted and consistent throughout the discussion.

Response: Lines 909–1192. We reviewed all references and adjusted the formatting for consistency where needed.

---

## [Decision Letter · Decision Letter 1]

18 Dec 2023

QTL analysis of femaleness in monoecious spinach and fine mapping of a major QTL using an updated version of chromosome-scale pseudomolecules

PONE-D-23-25608R1

Dear Dr. Onodera,

We’re pleased to inform you that your manuscript has been judged scientifically suitable for publication and will be formally accepted for publication once it meets all outstanding technical requirements.

Kind regards,

Mojtaba Kordrostami, Ph.D.

Academic Editor

PLOS ONE

Additional Editor Comments (optional):

The manuscript can be accepted now

Reviewers' comments:

Reviewer's Responses to Questions

**Comments to the Author**

1. If the authors have adequately addressed your comments raised in a previous round of review and you feel that this manuscript is now acceptable for publication, you may indicate that here to bypass the “Comments to the Author” section, enter your conflict of interest statement in the “Confidential to Editor” section, and submit your "Accept" recommendation.

Reviewer #1: All comments have been addressed

2. Is the manuscript technically sound, and do the data support the conclusions?

Reviewer #1: Partly

3. Has the statistical analysis been performed appropriately and rigorously? 

Reviewer #1: Yes

4. Have the authors made all data underlying the findings in their manuscript fully available?

Reviewer #1: Yes

5. Is the manuscript presented in an intelligible fashion and written in standard English?

Reviewer #1: Yes

6. Review Comments to the Author

Reviewer #1: (No Response)

7. PLOS authors have the option to publish the peer review history of their article (what does this mean?). If published, this will include your full peer review and any attached files.

Reviewer #1: **Yes: **Hamid Hatami Maleki

---

## [Editor Report · Acceptance letter]

16 Feb 2024

PONE-D-23-25608R1 

PLOS ONE

Dear Dr. Onodera, 

I'm pleased to inform you that your manuscript has been deemed suitable for publication in PLOS ONE. Congratulations! Your manuscript is now being handed over to our production team.

Kind regards, 

on behalf of

Dr. Mojtaba Kordrostami 

Academic Editor

PLOS ONE